# On the nature of hydrogen bonding in the H₂S dimer

Svenja Jäger [1,5], Jai Khatri [1,5], Philipp Meyer [1], Stefan Henkel[1], Gerhard Schwaab [1], Apurba Nandi [2], Priyanka Pandey [2], Kayleigh R. Barlow[3], Morgan A. Perkins[3], Gregory S. Tschumper [3], Joel M. Bowman [2] ✉, Ad van der Avoird [4] ✉ & Martina Havenith [1] ✉

Hydrogen bonding is a central concept in chemistry and biochemistry, and so it continues to attract intense study. Here, we examine hydrogen bonding in the H₂S dimer, in comparison with the well-studied water dimer, in unprecedented detail. We record a mass-selected IR spectrum of the H₂S dimer in superfluid helium nanodroplets. We are able to resolve a rotational substructure in each of the three distinct bands and, based on it, assign these to vibration-rotation-tunneling transitions of a single intramolecular vibration. With the use of high-level potential and dipole-moment surfaces we compute the vibration-rotation-tunneling dynamics and far-infrared spectrum with rigorous quantum methods. Intramolecular mode Vibrational Self-Consistent-Field and Configuration-Interaction calculations provide the frequencies and intensities of the four SH-stretch modes, with a focus on the most intense, the donor bound SH mode which yields the experimentally observed bands. We show that the intermolecular modes in the H₂S dimer are substantially more delocalized and more strongly mixed than in the water dimer. The less directional nature of the hydrogen bonding can be quantified in terms of weaker electrostatic and more important dispersion interactions. The present study reconciles all previous spectroscopic data, and serves as a sensitive test for the potential and dipole-moment surfaces.

After many years of intensive experimental and theoretical studies, the water dimer is now a textbook example of hydrogen bonding. Not only its structure with a near-linear hydrogen bond and its binding energy are important, but also the observation that it has eight equivalent hydrogen-bonded equilibrium structures separated by relatively small energy barriers. This implies that the dimer can quantum-mechanically tunnel between the corresponding minima in the potential energy surface (PES). The tunneling frequencies of H₂O dimers and larger clusters between multiple equivalent equilibrium structures, as well as

the frequencies of various intermolecular vibrations, have been the subject of numerous experimental studies by high-resolution molecular beam spectroscopy[1–6]. These tunneling and vibrational frequencies can also be accurately computed from a given intermolecular potential surface; they depend very sensitively on its shape[7–9]. Therefore, comparison with the measured data provides a very critical check of the quality of ab initio calculated water pair potentials[9–16]. The pair potential is the leading term in a many-body potential for liquid water and ice and accounts for 80–90% of the interactions in the bulk phase,

[1]Department of Physical Chemistry II, Ruhr University Bochum, 44801 Bochum, Germany. [2]Department of Chemistry and Cherry L. Emerson Center for Scientific Computation, Emory University, Atlanta, GA 30322, USA. [3]Department of Chemistry and Biochemistry, University of Mississippi, University, MS 38677-1848, USA. [4]Theoretical Chemistry, Institute for Molecules and Materials, Radboud University, Heyendaalseweg 135, 6525 AJ Nijmegen, Netherlands. [5]These authors contributed equally: Svenja Jäger, Jai Khatri. This paper is dedicated to the memory of Harold Linnartz and Giacinto Scoles. ✉e-mail: jmbowma@emory.edu; A.vanderAvoird@theochem.ru.nl; martina.havenith@rub.de

which triggered the extensive search for precise intermolecular potential energy surfaces[17]. Previously, we reported the low-frequency spectrum of water dimer in helium nanodroplets. This spectrum served as a sensitive test of the water dimer potential and dipole moment surfaces[18].

Sulfhydryl (SH) compounds are well known for their weak non-covalent interactions. Cysteine, for instance, plays a role in establishing side chain conformations and stability of secondary structures in peptides through inter- and intramolecular interactions[19–21]. The simplest SH-containing compound, hydrogen sulfide ($H_2S$), acts as a biological signaling molecule[22–24] and as a superconductor precursor[25,26]. $H_2S$ is the simplest sulfur-bearing molecule detected in the interstellar medium (ISM) and plays an important role in astrochemistry[27,28]. However, there remains a significant knowledge gap compared to its isovalent water ($H_2O$). Despite the significance of studying the pair interaction potential of $H_2S$ in detail, so far no high-quality potentials for $(H_2S)_2$ were available. The few IR studies reported so far, mostly in cryogenic matrices, were suffering from the overlap of spectra with higher cluster sizes and the small transition dipole moment of the dimer. In previous molecular beam studies[29], four broad IR bands have been reported in the frequency range between 2590 and 2620 $cm^{-1}$.

However, only a single vibrational transition in $(H_2S)_2$ is predicted to be intensive enough for observation in this frequency range. Thus, the assignment of the IR spectra was up to now a puzzle to be solved. Here, we present an intermolecular potential surface and use it to predict the different tunneling energy levels of the $H_2S$ dimer as well as its far-infrared spectrum, and compare them to those of the water dimer. We used our helium nanodroplet machine in Bochum to record the first mass-selective measurement of IR spectra of small $H_2S$ clusters. Based on these measurements, we can unambiguously assign three bands to the $H_2S$ dimer, while in the same frequency range also trimer bands are observed. After increasing the signal-to-noise ratio, we were able to resolve a rotational substructure of each of the dimer bands. As explained in the Methods section, $H_2S$ (just as $H_2O$) has two different nuclear spin isomers, called para and ortho $H_2S$ (p$H_2S$ and o$H_2S$), which have different rotational states. Based on the observed nuclear spin multiplicitiess of alternating rotational levels, see the table in the Methods section, we could assign the observed IR absorption peaks at 2598.2, 2602.0 and 2604.2 $cm^{-1}$ to the donor bound S-H stretch mode in ortho-ortho, ortho-para, and para-para $H_2S$ dimers and additional bands to larger clusters. So far, only transitions in one of these species −ortho-para− had been observed in microwave studies[30], which provided the rotational constants of its lowest two states. In contrast to the tetrahedral structure of ice, in the condensed form of $H_2S$ each molecule is surrounded by twelve neighbors[31,32], indicating a more isotropic and less directional intermolecular potential. The binding energy $D_e$ was calculated to be about 7 kJ/mol[33], which is roughly one third of the $D_e$ of the water dimer, indicating a shallower potential energy surface.

Based on the structure, as determined by microwave spectroscopy[30] and subsequently supported by ab initio calculations[33], the $H_2S$ dimer is known to be hydrogen bonded. However, its hydrogen-bonded geometry shows differences from that of the $H_2O$ dimer, which can be understood by considering the monomer structures. The S-H bonds in $H_2S$ are nearly perpendicular, with an HSH angle of 92°, while the HOH angle in $H_2O$ is about 104°, which is nearly the tetrahedral angle. This implies that both the OH binding orbitals and the lone electron pairs in $H_2O$ can be regarded as $sp^3$ hybrids, while in $H_2S$ the nearly perpendicular SH bonds are formed with the S atom $p$ orbitals, and the lone pairs are $sp$ hybrids pointing in opposite directions perpendicular to the $H_2S$ plane. Hence, the acceptor plane in the $H_2S$ dimer is nearly perpendicular to the SH bond of the donor involved in the hydrogen bond, in contrast with the $H_2O$ dimer where the OH bonds in the acceptor and the donor OH bond point to the corners of a tetrahedron.

Previously, low-resolution IR spectroscopic studies on the $H_2S$ monomer and dimer have been reported in cryogenic matrices[34–40], as well as in molecular beams[29] and in liquids[41]. However, information on the intermolecular potential energy surface is lacking so far. Furthermore, the assignment of the IR transitions was ambiguous, due to the small IR transition dipole of $H_2S$ and the spectral overlap of the IR bands of $H_2S$ clusters (dimers, trimers, or oligomers) in low-temperature matrices.

Here, we report an IR spectrum of $H_2S$ dimers in superfluid helium droplets in the frequency range of the bound S−H stretch mode. We were able to resolve a rotational structure and to assign the three bands observed at 2598.2, 2602.0, and 2604.2 $cm^{-1}$ to a single vibrational band with resolved tunneling states belonging to the ortho-ortho, ortho-para, and para-para nuclear spin isomers of the $H_2S$ dimer. This is in line with our calculations, which predict well separated energy levels for ortho-ortho, ortho-para, and para-para $H_2S$ dimers, but only a single strong vibrational transition in this frequency range. Bands observed at 2588 and 2620 $cm^{-1}$ were assigned to the trimer, based on the pick-up curves. Thus, we reassign the previously observed bands at 2590 and 2618 $cm^{-1}$ in molecular beams also to the trimer and the band at 2605 $cm^{-1}$ to the dimer.

Accompanying calculations based on a newly developed ab initio potential surface were carried out to predict the dimer vibration-rotation-tunneling (VRT) states. These take into account the large amplitude intermolecular vibrations of different molecular symmetries and allow us to predict also the far-infrared spectrum at 0.4 K, calculated with the new ab initio intermolecular potential and dipole function. The frequencies of the intermolecular vibrations are much lower than in the $H_2O$ dimer. Also, the tunneling splitting pattern of the VRT levels is very different: the donor-acceptor interchange splitting is much larger than in the water dimer, the acceptor switch splitting is substantially smaller. We discuss the nature of the various tunneling processes and of the large amplitude intermolecular vibrations in comparison with those of the water dimer. The zero-point energies of the three nuclear spin isomers are obtained from the VRT calculations, and the zero-point energy of the para-para species also from Diffusion Monte Carlo calculations in full dimensionality. From Vibrational Self-Consistent Field and Configuration Interaction[42,43] (VSCF/VCI) calculations we obtain the fundamental frequencies of the four SH-stretch modes, with a focus on the donor-bound SH mode which is observed experimentally. The calculations are in agreement with the conclusion from the experiment that the $a$-axis component of the transition intensity is the dominant one.

## Results

### Vibration-rotation-tunneling states

The lower bound vibration-rotation-tunneling VRT levels, calculated with the pseudospectral method are listed for p$H_2S$-p$H_2S$, o$H_2S$-o$H_2S$, and o$H_2S$-p$H_2S$ in Supplementary Tables 3, 4, 5. The tunneling levels are displayed graphically in Fig. 1 and the levels including intermolecular vibrations in Supplementary Fig. 2. The energy levels are characterized by the total angular momentum $J$, which is a good quantum number, and an approximate quantum number $K$, which is the projection of the total angular momentum $J$ on the intermolecular axis. We use the absolute value of $K$, because the VRT wave functions are even/odd combinations of functions with $+K$ and $-K$. From the energies with the same $K$ and $J = 0, 1, 2$ we derive the dimer end-over-end rotational constant $(B + C)/2$ and from the splitting between levels with the same $K > 0$ and different $\pm$ parity we extract $B − C$, see Table 1. If the dimer were a rigid rotor, the differences between the energies for different $K$ would yield the rotational constant $A$, but actually it is quite floppy, $K$ affects its internal motions, and thus $A$ is not a simple rotational constant.

The three tunneling processes that connect the eight equivalent global minima in the potential for water dimer are acceptor switch,

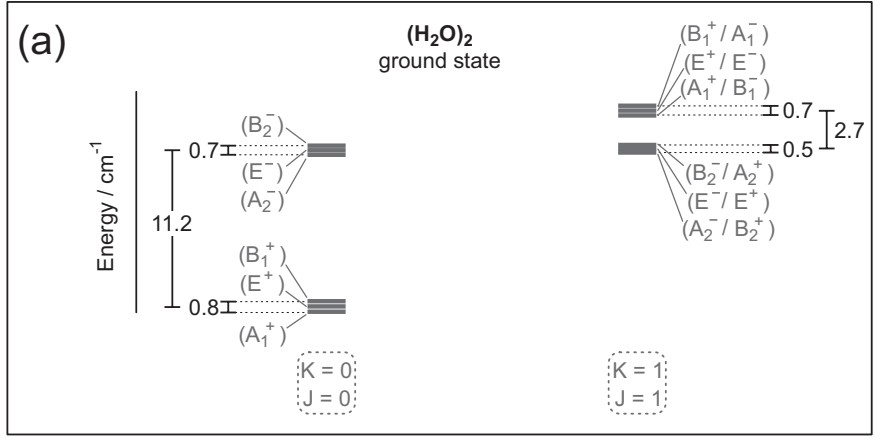

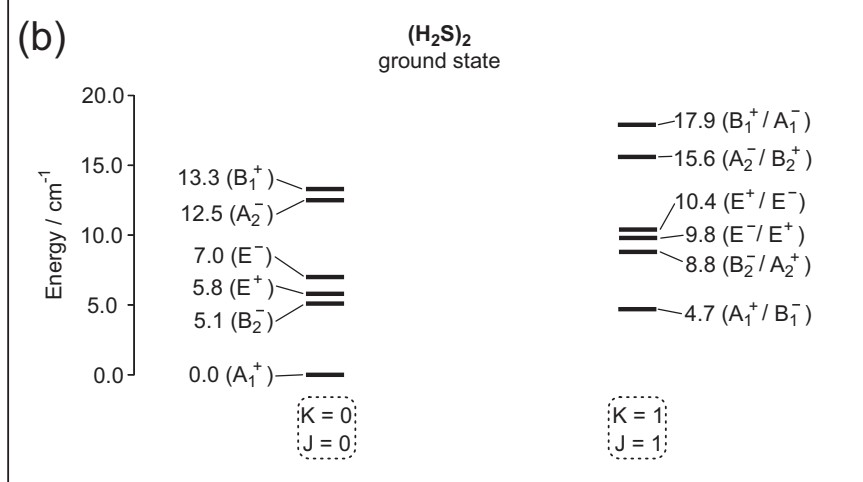

**Fig. 1 | Tunneling energy levels for J = K = 0 and J = K = 1.** (**a**) $H_2O$-$H_2O$ levels, (**b**) $H_2S$-$H_2S$ levels.

donor-acceptor interchange, and bifurcation (or donor) tunneling. For $H_2O$–$H_2O$ they give rise to the typical energy level pattern shown in Fig. 1a with a large acceptor tunneling splitting of about 11 cm⁻¹, considerably smaller splittings due to interchange, and very small shifts from bifurcation tunneling[8,9,18]. The corresponding picture for $H_2S$-$H_2S$ in Fig. 1b is very different, however. We can identify the tunneling processes in $H_2S$-$H_2S$ that give rise to this different picture by looking at the VRT wave functions in Supplementary Figs. 3 and 4. The acceptor switch tunneling splitting is 5.06 cm⁻¹, which is considerably smaller than the splitting of about 11 cm⁻¹ in $H_2O$-$H_2O$. This is probably related to the acceptor plane being nearly perpendicular to the hydrogen bond in $H_2S$-$H_2S$, which implies that the acceptor tunneling pathway is longer than in $H_2O$-$H_2O$. The interchange tunneling splitting, i.e., the energy difference between $A_1^+$ and $B_1^+$ is 13.29 cm⁻¹. This is much larger than the interchange splitting of about 0.7 cm⁻¹ in $H_2O$-$H_2O$, so the associated energy barrier is smaller. These differences

cause the energy level picture for $H_2S$-$H_2S$ to look very different from the corresponding picture for $H_2O$-$H_2O$.

The mixed o$H_2S$-p$H_2S$ states with symmetries $E^{\pm}$ are the only states that have a non-zero dipole moment because of the asymmetry in their wave functions, see Supplementary Fig. 5. The $A_{1,2}^{\pm}$ and $B_{1,2}^{\pm}$ states of p$H_2S$-p$H_2S$ and o$H_2S$-o$H_2S$ are symmetric or antisymmetric under $P_{AB}$, they all have equal weights for donor-acceptor and acceptor-donor structures, and their dipole moment averages to zero. Therefore, in the microwave spectrum[30] of $H_2S$-$H_2S$ one can only observe the transitions from $E^+$ to $E^-$ states and vice versa. In Table 1, we list our calculated rotational constants for the $E^+$ and $E^-$ states, together with the measured data[30]. We can now identify the lower and upper states mentioned in ref. 30 as the $E^+$ and $E^-$ states, respectively, which are separated in energy by 1.23 cm⁻¹. The maximum deviation of our calculated rotational constants from the values extracted from the microwave spectrum[30] is 1.27 %. This confirms the accuracy of the $H_2S$-$H_2S$ potential used in our VRT calculations.

In the $H_2O$ dimer, there is just one excited intermolecular vibrational state with energy below 100 cm⁻¹, while $H_2S$-$H_2S$ has many excited states below 65 cm⁻¹, see Supplementary Fig. 2 and Supplementary Tables 3, 4, 5. This illustrates that the $H_2S$ dimer is much floppier than the $H_2O$ dimer. The wave functions shown in Supplementary Fig. 6 have nodal planes that depend on at least two intermolecular coordinates. Due to the strong coupling of these coordinates the excited states are mostly of mixed character.

It is well known and illustrated by our results below that the vibrational frequencies of molecular dimers in He nanodroplets agree with the gas phase values to within a few cm⁻¹. While rotational levels of

**Table 1 | Rotational constants of the lowest $E^+$ and $E^-$ states, corresponding to o$H_2S$-p$H_2S$**

|  | Lower state ($E^+$) | | Upper state ($E^-$) | |
|---|---|---|---|---|
|  | Calculated | Experimental[30] | Calculated | Experimental[30] |
| A(MHz) | 116400.4 |  | 97831.2 |  |
| B(MHz) | 1775.4 | 1753.1019(28) | 1761.9 | 1752.8788(11) |
| C(MHz) | 1752.0 | 1743.1163(28) | 1742.4 | 1745.7388(11) |
| $d_J$(kHz) | 16.7 | 15.227(11) | 10.9 | 14.921(11) |

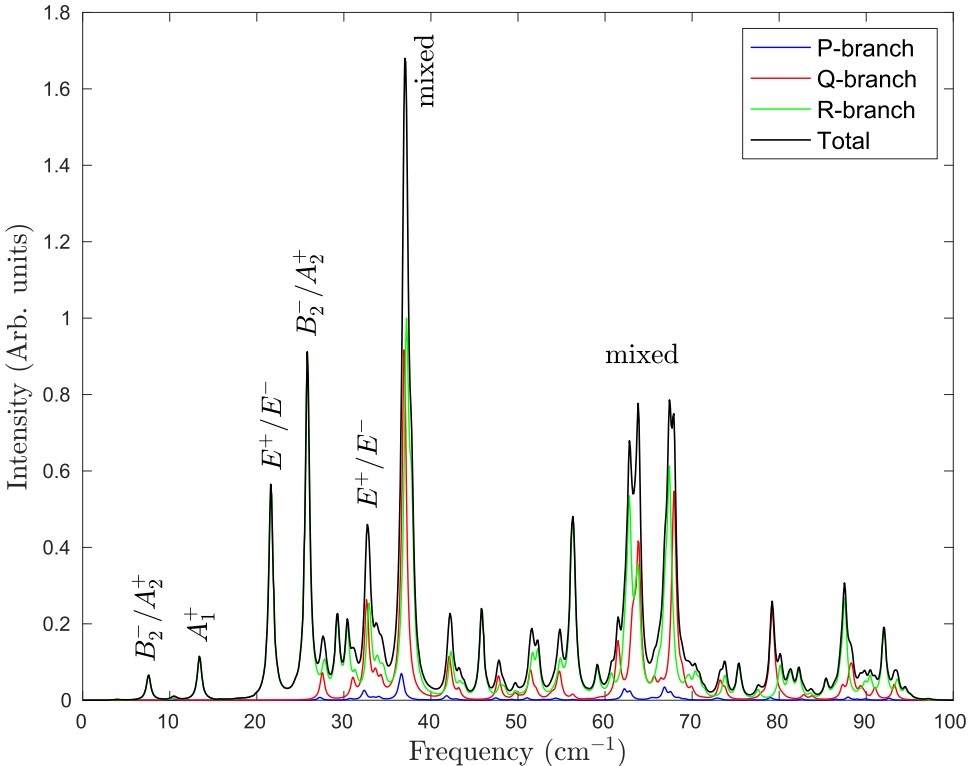

**Fig. 2 | Far-infrared spectrum of H₂S-H₂S at _T_ = 0.37 K calculated with the ab initio intermolecular potential and dipole function.** The lines in the spectrum were assigned with the use of the transition frequencies and line strengths calculated separately for each of the symmetries.

larger complexes with small energy splittings can couple to the phonon bath of helium droplets larger energy spacings are mostly unaffected. For a direct comparison of experimentally observed intermolecular modes in the gas phase and in helium droplets, we refer to our papers on water dimer and water trimer[18,44], with the frequencies of the predicted and measured intermolecular modes agreeing to within the experimental uncertainty.

### Theoretical far-infrared spectrum

The far-IR spectrum of H₂S-H₂S at temperature $T$ = 0.37 K, which is the temperature of He droplets in the molecular beam, is shown in Fig. 2. It is calculated with the Lanczos-based recursive residue generator method[45,46] and the six-dimensional (6D) dipole function described in the Methods section below. Due to nuclear spin conservation, the lowest states of each permutational symmetry are populated. Since dipole transitions are only allowed between states with the same permutational

### Table 2 | Assignment of the lines in the theoretical far-infrared spectrum at _T_ = 0.37 K

| Frequency | Lower state | | Nature of excited state |
|---|---|---|---|
| (cm⁻¹) | even _J_ | odd _J_ | |
| 7.5 | $B_2^-$ | $A_2^+$ | acceptor tunneling |
| 13 | $A_1^+$ | $B_1^-$ | interchange tunneling |
| 22 | $E^+$ | $E^-$ | donor stretch / in − plane bend |
| 26 | $B_2^-$ | $A_2^+$ | donor torsion / out − of − plane bend |
| 32 | $E^+(K=0)$ | $E^-(K=0)$ | $E^-$, $E^+(K=1)$ |
| 37 | various | various | mixed |

The symmetry labels are defined in the table in the Methods section. States with symmetry $B_1^-$ have weight zero, so these odd _J_ states are missing.

symmetry and opposite parity, the spectrum contains contributions from transitions starting from the occupied states for each of the symmetries with weights given by their nuclear spin multiplicities.

The lines in the spectrum of Fig. 2 were assigned with the use of the transition frequencies and line strengths calculated for each of the symmetries with the analytical method. The assignments are indicated in Fig. 2 and are listed in Table 2. The higher peaks are not specified, because the corresponding excited states are more and more complex.

The experimental mid-infrared spectrum of H₂S-H₂S presented in this paper involves excitation of the bound S-H stretch mode in the donor. Although the dimer with one internally excited monomer formally has the same permutation-inversion symmetry group $G_{16}$, either monomer can be excited and the number of dimer excited states is larger than the number of ground states. Effectively, states in which the donor is excited differ from those in which the acceptor is excited, but these states are connected by interchange tunneling. However, the interchange tunneling splitting is expected to be substantially smaller than in the ground state, because the equivalence of excited donor-acceptor and acceptor-donor states not only requires a geometry change but also hopping of the excitation from one monomer to the other. Both measurements and calculations on HF and HCl dimers in their ground and monomer excited states[47–55] support this assumption.

Although the explicit calculation of the monomer excited states of H₂S-H₂S would require 12-dimensional (12D) calculations, as reported[15,56] for H₂O-H₂O, we can give some indications on the basis of our 6D calculations that are useful to assign the measured mid-IR spectrum. At a temperature of $T$ = 0.37 K in the He droplets in which the dimer was embedded and equilibrated, only the lowest states of each symmetry are populated. Given the nuclear spin weights in the table in the Methods section, this implies that lines in the spectrum due to excitations from the lowest $B_2^-$ states with even _J_ and $A_2^+$ states with odd _J_ alternate in intensity since the nuclear spin weights of these states are 3 and 6, respectively. Such an intensity alternation will not occur for excitations from the lowest $E^+$ states with even _J_ and $E^-$ states

with odd $J$ which have the same nuclear spin weight 3. For the lowest $A_1^+$ states with even $J$ the situation is different again: the corresponding $B_1^-$ states with odd $J$ have nuclear spin weight zero. This implies that transitions from these lowest $A_1^+$ states are only allowed for even initial $J = 0, 2$, etc.

## Dissociation energy

The dissociation energy, $D_0$, is obtained using an extrapolated value of $D_e$ of 583 cm$^{-1}$ at the CCSD(T) complete basis set (CBS) limit (details in Supplementary Table 6), plus twice the zero-point energy (ZPE) of the isolated H$_2$S monomer minus the zero-point energy of the H$_2$S dimer. The latter is obtained from standard unconstrained diffusion Monte Carlo (DMC) calculations, using the present 12D PES, following the protocol used previously for the calculation of $D_0$ for the water dimer[57]. The resulting ZPE of the H$_2$S dimer is 6822 cm$^{-1}$ with an uncertainty of several cm$^{-1}$. The ZPE of the monomer is taken from the literature[58], which reported exact variational calculations on an accurate ab initio PES, which yield excellent agreement with experimental values for vibrational transitions. The H$_2$S ZPE reported there is 3294.3 cm$^{-1}$ and thus for two monomers the ZPE is 6588.6 cm$^{-1}$. From these numbers we obtain 349 cm$^{-1}$ for $D_0$ with a conservative estimated uncertainty of 10 cm$^{-1}$. Note this $D_0$ corresponds to total angular momentum $J = 0$ of the dimer and in DMC, the wave functions are assumed to be nodeless, so this $D_0$ value refers to pH$_2$S-pH$_2$S.

As already pointed out, three different dimer nuclear spin isomers exist: pH$_2$S-pH$_2$S, oH$_2$S-pH$_2$S, and oH$_2$S-oH$_2$S. They have different $D_0$ values because nuclear spin is conserved upon dissociation, and they dissociate into the corresponding para and ortho monomers. From CCSD(T) computations near the CBS limit with the rigid-monomer geometries used in the 6D calculations of the VRT states it is found that $D_e = 579$ cm$^{-1}$. In order to obtain the $D_0$ values, one needs the ZPE's of each of the three species given by its lowest VRT level, as well as the energies of the lowest allowed rotational states of the para and ortho monomers.

The ZPE's of pH$_2$S-pH$_2$S, oH$_2$S-pH$_2$S, and oH$_2$S-oH$_2$S given in Supplementary Tables 3, 4, and 5 are 241, 247, and 246 cm$^{-1}$, respectively. The ground rotational state of pH$_2$S is the $j_{k_a k_c} = 0_{00}$ state with energy zero and with experimental H$_2$S rotational constants[59] one finds that the lowest allowed rotational state $1_{01}$ of oH$_2$S has an energy of 13.7491 cm$^{-1}$. The resulting $D_0$ values of pH$_2$S-pH$_2$S, oH$_2$S-pH$_2$S, and oH$_2$S-oH$_2$S are 338, 346 and 361 cm$^{-1}$, respectively. The value for pH$_2$S-pH$_2$S agrees with the value from DMC calculations on the 12D potential to about the estimated statistical uncertainty of 10 cm$^{-1}$ in the latter value.

## Experimental Results

In Fig. 3, we show the experimentally observed IR spectrum of H$_2$S clusters in helium nanodroplets in the frequency range of the bound

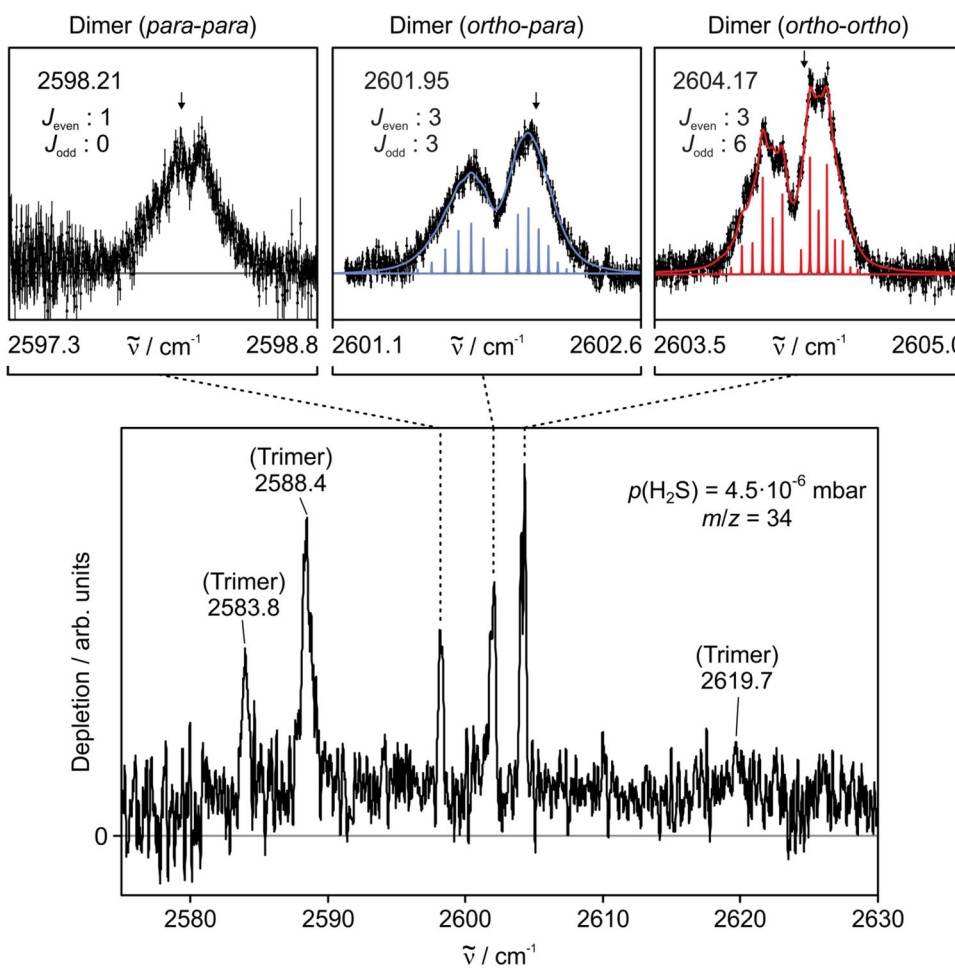

**Fig. 3 | Experimentally recorded IR spectrum of H$_2$S clusters in helium nanodroplets at 0.37 K, $m/z$ = 34 and partial H$_2$S pressure of 4.5 × 10$^{-6}$ mbar.** We observe three bands of H$_2$S-H$_2$S centered at 2598.21, 2601.95, and 2604.17 cm$^{-1}$, which could be assigned to three different vibrational-rotational-tunneling transitions. In the zoom, we show a comparison of the experimentally recorded rotational line shape (black) with the adapted predicted spectrum (colored sticks) assuming alternating or non-alternating intensity ratios of subsequent rotational levels according to the respective spin statistics. Black arrows show the wavenumber where pick-up curves were recorded (see Supplementary Information).

donor stretch, i.e., between 2570 cm$^{-1}$ and 2630 cm$^{-1}$ at the mass-to-charge ratio $m/z = 34$ corresponding to $(H_2^{32}S)^+$. IR transitions are observed at 2583.8, 2588.4, 2598.2, 2602.0, 2604.2, and 2619.7 cm$^{-1}$. At each of these peak frequencies, we recorded so-called pick-up curves for assignment to a specific cluster, see Supplementary Note 6 for more details. Based on these pickup curves, the bands at 2598.2, 2602.0, and 2604.2 cm$^{-1}$ could be assigned to H$_2$S dimers, while the bands at 2583.8, 2588.4, and 2619.7 cm$^{-1}$ are attributed to H$_2$S trimers. In the dimer bands we clearly observe $P$ and $R$ branches. The substructure of these branches could be reproduced to experimental uncertainty when taking into account the correct spin statistics: alternating spin weights (3:6) for subsequent $J$ levels for the ortho-ortho transition, (3:3) for the ortho-para transition, and (1:0) for the para-para transition. We were unable to observe any monomer bands, even at lower H$_2$S pressures. Note that also in the gas phase, only weak bands have been reported previously at 2615 and 2625 cm$^{-1}$ [41].

In Table 3 we present harmonic and VSCF/VCI excitation energies and intensities for the indicated four S-H stretches. Given the strong

evidence from calculations that the wave functions sample the three low-energy minima, we present results that are averaged over these minima along with standard deviations. The corresponding structures labeled Minimum I, II, and III are shown in Supplementary Fig. 7.

The first indications for feasible tunneling were found in the microwave spectrum of the H$_2$S dimer measured by Das et al.[30] in a molecular beam. They reported rotational transitions from two distinct ground states. These two states are the $E^+$ and $E^-$ states, which we predict to be separated by 1.23 cm$^{-1}$. Using a Boltzmann distribution it can be estimated that at the molecular beam temperature of 4 K the populations of the $E^+$ and $E^-$ levels are 0.61 and 0.39, whereas at the superfluid helium droplet temperature of 0.37 K these are 0.992 and 0.008. Therefore, only the $E^+$ state is expected to be populated in superfluid helium droplets.

While in previous studies[29,35–37,39] the IR bands observed in molecular beams and in matrices were assigned to the $v_1$ acceptor and $v_3$ donor/acceptor modes, all observed H$_2$S dimer bands are now assigned to ro-vibrational tunneling transitions involving the bound donor SH stretch mode, see Table 4. If we assume that all transition dipole moments are nearly the same, the intensity ratio of the various lines in the spectrum is determined by the nuclear spin multiplicities given in the table in the Methods section. We propose the following assignments: $A_1^-(J_{odd}, K_a = 0) \leftarrow A_1^+(J_{even}, K_a = 0)$ at 2598.2 cm$^{-1}$, $E^-(J_{odd})/E^+(J_{even}), (K_a = 0) \leftarrow E^+(J_{even})/E^-(J_{odd}), (K_a = 0)$ at 2602.0 cm$^{-1}$, and $B_2^+(J_{odd})/A_2^-(J_{even}), K_a = 0 \leftarrow B_2^-(J_{even})/A_2^+(J_{odd}), K_a = 0$ at 2604.2 cm$^{-1}$.

To ensure an unambiguous assignment of the IR spectra we tested different spin statistical weight ratios of the rotational lines in the $P$ and $R$ branches of the band at 2604.2 cm$^{-1}$. It was found that a spin statistical weight ratio of 3:6 for $J_{even}$:$J_{odd}$ yields the smallest mean square deviation between measurement and fit. As explained in the last paragraph of the section "Theoretical far-infrared spectrum", this intensity ratio of 3:6 indicates that this band belongs to oH$_2$S-oH$_2$S with the lower states in the transitions having $B_2^-$ symmetry for even $J$ and $A_2^+$ symmetry for odd $J$. The fitted rotational constant of this band is $B = (0.0259 \pm 0.0003)$ cm$^{-1}$ and the difference between the vibrational ground and excited state values is $\Delta B = (-0.0009 \pm 0.0001)$ cm$^{-1}$. In the band at 2602.0 cm$^{-1}$ no intensity alternation was observed. This agrees with an intensity ratio of 3:3, indicating that this band belongs to oH$_2$S-pH$_2$S, with lower states of $E^+$ symmetry for even $J$ and $E^-$ symmetry for odd $J$. The fitted rotational constants of this band are $B = (0.02230 \pm 0.0001)$ cm$^{-1}$ and $\Delta B = (-0.00067 \pm 0.00005)$ cm$^{-1}$. No unambiguous fit of the band at 2598.2 cm$^{-1}$ was possible due to the low signal-to-noise ratio. The nearest lines in the $P$ and $R$ branches of this band are separated by 0.14 cm$^{-1}$, while for the other two bands, we find gaps of 0.09 and 0.10 cm$^{-1}$. This is in agreement with an assignment of the 2598.2 cm$^{-1}$ band to pH$_2$S-pH$_2$S, with lower levels of $A_1$ symmetry

**Table 3 | VSCF/VCI frequencies (cm$^{-1}$), intensities (arbitrary units) and percent contributions from $a$, $b$, and $c$ principal axis components for different SH-stretch modes with symmetries $A'$ and $A''$ for Minimum I, II, and III of the H$_2$S dimer**

| Minimum | Mode | Freq. | a | b | c | Intensity |
|---|---|---|---|---|---|---|
| **I** | Bound donor($A'$) | 2578 | 99.9 | 0.0 | 0.12 | 5.339 |
| | Symmetric acceptor stretch($A'$) | 2612 | 63.5 | 0.0 | 36.4 | 0.154 |
| | Free donor stretch($A'$) | 2632 | 90.3 | 4.5 | 5.2 | 0.212 |
| | Asymmetric acceptor stretch($A''$) | 2633 | 2.1 | 97.7 | 0.1 | 0.298 |
| **II** | Bound donor($A'$) | 2610 | 0.4 | 0.0 | 99.6 | 0.247 |
| | Symmetric acceptor stretch($A'$) | 2623 | 96.1 | 0.0 | 3.9 | 0.201 |
| | Free donor stretch($A'$) | 2646 | 94.0 | 0.0 | 6.0 | 0.276 |
| | Asymmetric acceptor stretch($A''$) | 2648 | 0.0 | 100.0 | 0.0 | 0.049 |
| **III** | Bound donor($A'$) | 2583 | 94.6 | 0.0 | 5.4 | 5.150 |
| | Symmetric acceptor stretch($A'$) | 2611 | 72.4 | 0.0 | 27.6 | 0.143 |
| | Free donor stretch($A'$) | 2624 | 76.9 | 0.0 | 23.1 | 0.423 |
| | Asymmetric acceptor stretch($A''$) | 2634 | 0.0 | 100.0 | 0.0 | 0.219 |

See text for more details.

**Table 4 | Observed bands (frequencies in cm$^{-1}$) of H$_2$S clusters in helium nanodroplets (our measurements), in molecular beams, and in matrices**

| Our assignment | He droplets | Previous[29] assignment | Molecular beam[29] | Ne[35] matrix | Ar[36] matrix | Kr[37]/Xe[38] matrix | N$_2$[36,39] matrix | CO[36] matrix |
|---|---|---|---|---|---|---|---|---|
| Trimer | 2588.4 | $v_1$ donor | 2590 | 2596.5 | 2569.5 | Kr: 2577.0 | 2580.3 | 2567 |
| | | (dimer) | (100) | (s) | (s) | (71) | (s) | (s) |
| | | | | | | Xe: 2572.5 | | |
| $v_1$ donor | 2598.2 | $v_1$ acceptor | 2605 | 2605.0 | 2585.5 | Kr: 2600.8 | 2617.8 | 2604 |
| (dimer) | 2602.0 | (dimer) | | (25) | (m) | (8) | (w) | (w) |
| | 2604.2 | | | | | | | |
| Trimer | 2619.7 | $v_3$ donor | 2618 | 2622.1 | 2624.3 | Kr: 2629.2 | 2625.3 | 2614 |
| | | (dimer) | | (10) | (w) | (9) | (w) | (w) |
| Trimer | 2619.7 | $v_3$ acceptor | 2618 | 2622.1 | 2629.1 | Kr: 2614.2 | 2631.1 | 2618 |
| | | (dimer) | | (10) | (w) | (9) | (w) | (w) |

Numbers in parentheses are the reported IR intensities. The frequencies from our VSCF/VCI calculations are 2578, 2610, and 2583 cm$^{-1}$ for Minimum I, II, and III of the H$_2$S dimer, respectively. The corresponding intensities in Table 3 show that only a single band is predicted to have sufficient intensity.

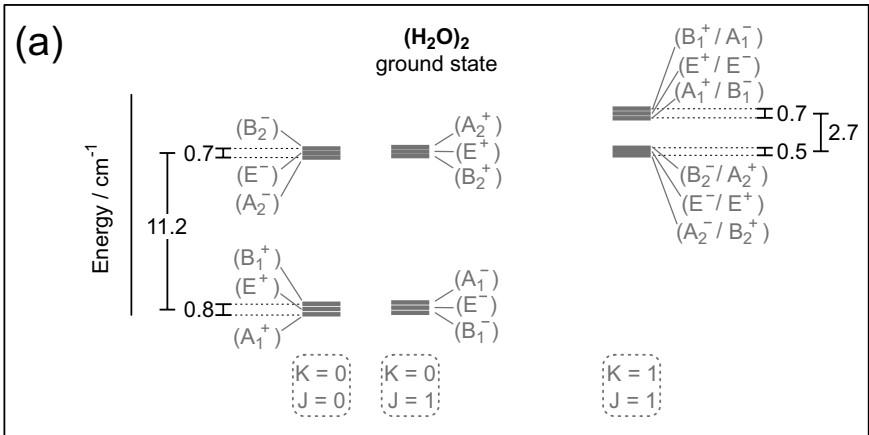

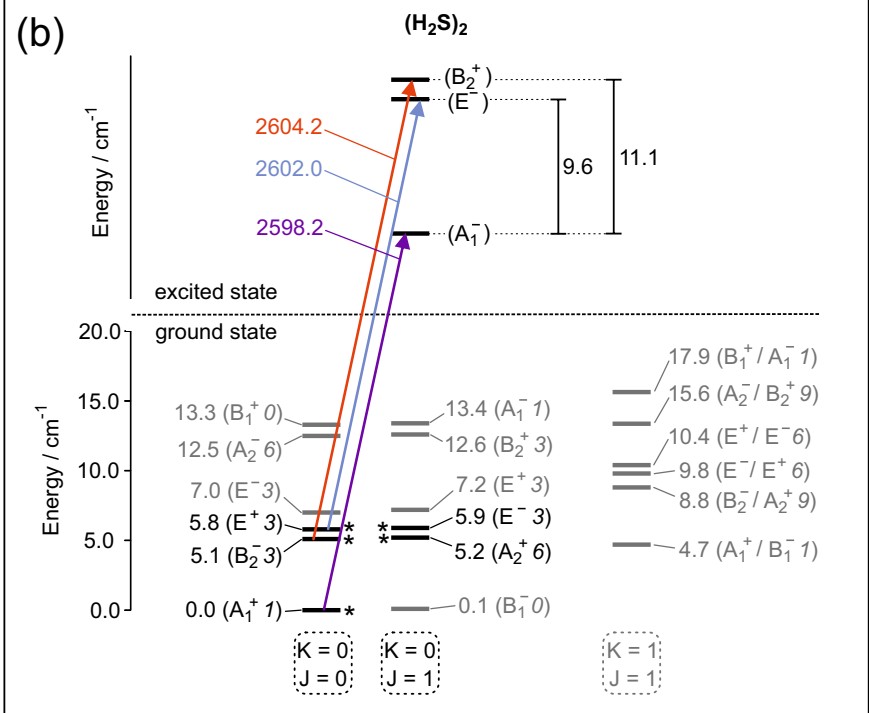

**Fig. 4 | Energy level schemes of the H₂O and H₂S dimers.** (a) H₂O dimer in the vibrational ground state, (b) H₂S dimer in the ground and vibrationally excited state. The energy scheme in the ground state corresponds to our VRT levels calculated on a high-quality ab initio intermolecular potential. The arrows represent experimentally observed bands. The stars mark the enery levels populated at 0.37 K.

for even $J$ and $B_1$ symmetry for odd $J$. Since $B_1$ states have spin statistical weight zero, see the table in the Methods section, the odd $J$ levels are missing. So the spacing should be $6B$, while for the ortho-ortho and ortho-para dimers it is $4B$, which nicely agrees with the observed gaps and confirms our assignment. The rotational constants in the helium droplets are smaller by a factor of 2.3 compared to the gas phase, which is in good agreement with previous studies[60].

In the vibrationally excited state, the energy level spacings are expected to differ from the ground state spacings, since upon the vibrational excitation of the donor, the barrier in any tunneling path involving donor-acceptor interchange is expected to be significantly increased. Based on the experimentally observed band center frequencies and the predicted tunneling splittings in the ground state, we can deduce the tunneling splittings in the vibrationally excited states shown in the energy level diagram in Fig. 4. Indeed, upon vibrational excitation, the donor-acceptor interchange splitting becomes much smaller. It is notable that the resulting splitting pattern in the energy

level scheme of the excited state of the H₂S dimer resembles remarkably well the tunneling splitting pattern in the ground state of the water dimer.

Table 3 gives calculated VSCF/VCI frequencies and intensities using the 12D PES at the three minima. Recall that these calculations are restricted to the six intramolecular modes, and do not reflect the delocalized nature of the exact ground vibrational wavefunction. The frequencies are ≈100 cm⁻¹ downshifted relative to the harmonic frequencies (given at Minimum I in Supplementary Table 2). VSCF/VCI stick spectra, shown in Supplementary Fig. 7, span a range from 1000 to 4000 cm⁻¹ and include combination and overtone transitions.

A summary of experimental results from a variety of experiments, including the present He nanodroplet one, is given in Table 4. From these, it can be concluded that the VSCF/VCI results are in good agreement with experiment, i.e., within about 10 cm⁻¹. Note the VSCF/VCI calculations are "vibrational only", i.e., for $J = 0$ and no attempt was made to determine the experimental energies of the triplet reported here.

## Discussion

The question whether the $H_2S$ dimer is hydrogen bonded and how its properties compare with those of its smaller sibling, the $H_2O$ dimer, attracted the interest of a broad chemical community[61,62]. It triggered several infrared spectroscopic studies of the $H_2S$ dimer in molecular beams[29] and in various rare gas and other matrices[36–40]. The spectra showed several partly overlapping bands in the frequency range of the S-H stretch mode, but their assignment remained uncertain due to the low resolution, the possible effects of different matrix sites, and the possibility that some of the bands belong to trimers and larger clusters. The microwave spectrum of the $H_2S$ dimer measured in a molecular beam[30] revealed clearly that this dimer has a hydrogen-bonded geometry, which evoked much attention[61,62]. A hydrogen bonded geometry was also found in subsequent ab initio calculations of the dimer's equilibrium structure[33].

The vast amount of studies of the $H_2O$ dimer not only addressed its structure, but also the different tunneling processes connecting eight equivalent equilibrium structures and the intermolecular vibrations. High-resolution far-infrared spectra[1–6], in combination with accurate calculations of its VRT states, yielded interesting information on the tunneling dynamics and the low-frequency intermolecular vibrations, which served as a very critical test of the available ab initio water pair potentials[9–16]. Such information was still lacking for the $H_2S$ dimer, but it is provided here. We present mass-selected IR spectra of the $H_2S$ dimer in helium nanodroplets in a molecular beam, accompanied by calculations of its VRT states with the use of a new accurate ab initio pair potential and dipole function. The low temperature of 0.37 K and the superfluid nature of the helium droplets made it possible to resolve and analyze the rotational substructure of the observed infrared bands and, with the help of the calculated tunneling splittings in the vibrational ground state and the symmetry relations, to unambiguously assign the three bands in the measured spectrum. All three experimentally observed dimer bands are attributed to the stretch mode of the donor SH group directly involved in the hydrogen bond; this assignment is supported by VSCF/VCI calculations on the intramolecular vibrations which show that only this mode has sufficient intensity to be observed. Just as $H_2O$, $H_2S$ has two nuclear spin isomers, ortho and para, which have different rotational states. The splitting of the donor SH stretch transition into three distinct bands is caused by different tunneling splittings in ortho-ortho, ortho-para, and para-para $H_2S$ dimers. Also the different dissociation energies $D_0$ of the ortho-ortho, ortho-para, and para-para $H_2S$ dimers into the corresponding monomers are reported here. The microwave spectrum[30] only addressed the mixed ortho-para dimer, because it is the only dimer species with a non-vanishing dipole moment. The rotational constants calculated for the lowest two tunneling states of this ortho-para dimer agree well with the values deduced from the microwave spectrum[30], which confirms that our ab initio $H_2S$ dimer potential is accurate.

The calculations presented provide a full 12-dimensional potential and dipole function of the $H_2S$ dimer depending on all intra- and intermolecular degrees of freedom. DMC calculations on this 12D potential yield the vibrational ground state of the dimer, while rigorous variational quantum calculations on the 6-dimensional intermolecular potential yield the tunneling and low-frequency intermolecular vibrational states. Moreover, we predict the $H_2S$ dimer's far-infrared spectrum, with the use of the 6D dipole function.

The tunneling splitting pattern in the $H_2S$ dimer is found to be very different from the well-known pattern in the $H_2O$ dimer, which is mainly caused by more facile donor-acceptor interchange tunneling and more strongly hindered acceptor switch tunneling. Also the intermolecular vibrations differ strongly from those of the $H_2O$ dimer, which is reflected by the predicted far-infrared spectrum. They have much lower frequencies, larger amplitudes, and different intermolecular modes are more strongly mixed. All of this indicates that the $H_2S$ dimer is much floppier than the $H_2O$ dimer. Calculations by

symmetry-adapted perturbation theory, see Supplementary Table 7, indicate that the less directional character of the hydrogen bond in the $H_2S$ dimer, as compared to the $H_2O$ dimer, is caused by a less dominant role of the electrostatic dipole and quadrupole interactions and relatively larger dispersion interactions, associated with the larger polarizability of $H_2S$. In summary, we conclude that the experimental spectrum and its analysis, and the calculations presented in this paper clearly show that the $H_2S$ dimer, although hydrogen bonded, shows very different dynamical properties than the isovalent $H_2O$ dimer. The insights gained in this study of the $H_2S$ dimer may also lead to a better understanding of the nature of non-covalent bonds involving sulfhydryl (SH) groups in biological systems, as compared to bonds involving OH groups.

## Methods

### Experimental details

The Bochum helium nanodroplet machine has been described in detail before[63,64]. It consists of five differentially pumped vacuum chambers interconnected by conflate flanges. Helium droplets were generated in the first chamber (expansion chamber), where pressurized helium gas passes through the precooled nozzle of 5 micron diameter into vacuum. The nozzle was operated at 10–20 K, and a backing pressure of 40–50 bar to form the different sizes of droplets (1000–10,000 He atoms). The pressure in the expansion chamber is kept at about $10^{-5}$ mbar. The helium droplets are collimated by a skimmer of 0.5 mm diameter before they enter the first pickup chamber which is kept at a pressure of $10^{-6}$ mbar; with a partial pressure of $6.5 \times 10^{-6}$ mbar for pick up of $H_2S$. The last chamber (detection chamber) hosts a quadrupole mass spectrometer (Pfeiffer QMG 422) to ionize the droplet beams. The mass spectrometer has two alternative modes for detection, either mass selective (at a given $m/z$) or via a high pass filter (detection of all ionized helium clusters and fragments with $m/z > 4$) The pressure in the detection chamber was kept in the range of $10^{-10}$ mbar. As a radiation source served a continuous wave optical parameter oscillator (OPO, Lockheed-Martin Aculight ARGOS 2400SF15 Module C). A 15 mW Yb-doped fiber laser at 1064 nm, which was amplified to 15 W using a diode-pumped fiber, served as a seed laser. The pump laser is split into an idler and a signal beam. The OPO laser source can be scanned in the wave length range between 3.2 $\mu m$ and 3.9 $\mu m$, the average idler output power is 1 W, the frequency resolution is 30 MHz. A Bristol 621A wavemeter is used for frequency calibration. For phase-sensitive detection, the laser was chopped with a frequency of 25-35 Hz before entering the detection chamber.

### Theory

**Potential and dipole moment surfaces.** Here we report a new full twelve-dimensional (12D) potential energy surface that makes use of Δ-Machine Learning to bring an MP2-based PES to near CCSD(T) quality, using a limited number of explicitly correlated CCSD(T) energies[65,66]. In the present case, these are CCSD(T)-F12b/haQZ-F12 calculations, as described in detail in Supplementary Note 1. In addition, an MP2-based dipole moment surface (DMS) is reported. Both the PES and DMS are represented by permutationally invariant polynomials[67,68]. The new PES and DMS are used in diffusion Monte Carlo calculations of the ground vibrational state, extensive rigorous rigid-monomer quantum calculations of the VRT states and transition line intensities, and approximate VSCF/VCI calculations for intramolecular mode excitations.

**Rigid-monomer quantum calculations.** The rigid-monomer calculations of the VRT states use a 6D intermolecular potential obtained from the full 12D potential by fixing the geometry of both $H_2S$ monomers at an SH bond length of 1.3384 Å and an HSH angle of 92.5°. This monomer geometry is close to the geometry of the hydrogen bond acceptor, which is the least perturbed monomer, at the global minimum of the 12D potential for the $H_2S$ dimer. It must be the same for

**Table 5 | Permutation-inversion group $G_{16}$, irrep definitions**

|          | monomers   | $P_{AB}$ | $E^*$ | multiplicity |
|----------|------------|----------|-------|--------------|
| $A_1^+$  | para-para  | even     | +     | 1            |
| $A_1^-$  | para-para  | even     | −     | 1            |
| $B_1^+$  | para-para  | odd      | +     | 0            |
| $B_1^-$  | para-para  | odd      | −     | 0            |
| $A_2^+$  | ortho-ortho| even     | +     | 6            |
| $A_2^-$  | ortho-ortho| even     | −     | 6            |
| $B_2^+$  | ortho-ortho| odd      | +     | 3            |
| $B_2^-$  | ortho-ortho| odd      | −     | 3            |
| $E^+$    | ortho-para |          | +     | 3            |
| $E^-$    | ortho-para |          | −     | 3            |

$P_{AB}$ is the interchange permutation, $E^*$ is the inversion operation, the last column gives the nuclear spin multiplicities.

both monomers in order to reproduce all tunneling effects including donor-acceptor interchange. The 6D intermolecular potential depends on the monomer-monomer center-of-mass distance $R$ and five Euler angles defining the orientations of the monomers. The VRT states of $H_2S$-$H_2S$ are calculated with two different methods, which were applied earlier to the $H_2O$ dimer[7,9]. The first method is the split Wigner pseudospectral approach[7] that employs a grid basis in all six intermolecular coordinates in combination with an analytic basis. The second method[9] uses only the analytic angular basis and expands the 6D potential in angular expansion functions. Details are given in the Supplementary.

In addition, we predicted the far-infrared spectrum with two different methods. The first method uses recursive residue generation[45,46] to directly get the transition dipole moments as the projections of eigenstates multiplied with the dipole function onto the initial Lanczos vectors without actually computing all final eigenstates, which can be efficiently done in the grid basis of the split Wigner pseudospectral approach. The second method expands not only the potential but also the dipole moment function in angular basis functions and analytically calculates the transition dipole moment over the initial and final states. The dipole moment function in the six intermolecular coordinates is obtained from the 12D dipole moment surface in the same way as the 6D rigid-monomer intermolecular potential is derived from the 12D potential surface, with the same fixed $H_2S$ monomer geometry. The symmetry group of the $H_2S$ dimer in its equilibrium geometry is the point group $C_s$. But the dimer has eight equivalent equilibrium structures with rather low energy barriers between the corresponding global minima in the potential, so the molecular symmetry used in our calculations is the permutation-inversion (PI) or molecular symmetry[69] group $G_{16}$. This group has 10 irreducible representations (irreps) listed in Table 5. We note here that for the $H_2S$ monomer, just as for $H_2O$, one denotes rotational states with the asymmetric rotor quantum numbers $j_{k_a k_c}$. We used the most abundant $^{32}S$ isotope with nuclear spin $I = 0$. States that are even under the permutation $P_{12}$ that interchanges the two H atoms have even $k_a + k_c$, are called para-$H_2S$ (p$H_2S$), and have total nuclear spin $I = 0$ (multiplicity 1), and states odd under $P_{12}$ have odd $k_a + k_c$, are called ortho-$H_2S$ (o$H_2S$), and have total nuclear spin $I = 1$ (multiplicity 3). These monomer nuclear spin weights, combined with the permutation $P_{AB}$ that interchanges the monomers A and B, cause the $H_2S$ dimer states for different irreps to have the nuclear spin multiplicities given in Table 5. The symbol $E^*$ denotes the inversion operation. In both methods to compute the VRT states the calculations were performed separately for each irrep with a symmetry-adapted basis, as explained in Refs. 7,9.

The dipole moment function is invariant under all permutations, just as the Hamiltonian, but it changes sign under inversion. So for the overall vibration-rotation-tunneling transition it carries the irrep $A_1^-$.

Electric dipole transitions are only allowed between irreps with the same permutation symmetry and opposite ± parity under inversion.

**VSCF/VCI and diffusion Monte Carlo calculations.** The VSCF/VCI calculations employed the code MULTIMODE[70] and are not at the same level of rigor as the rigid-monomer quantum calculations described in the preceding section. They do not take into account the delocalized nature of the VRT states, but instead they are done at three reference configurations which are nearly isoenergetic. Moreover, they do not include rotation of the dimer ($J = 0$)[33]. The Diffusion Monte Carlo calculations of the zero-point energy in 12D are exactly as reported previously for the water dimer[57]. Details of these calculations are given in the Supplementary.

## Data availability
All data supporting the findings of this study are available within the article and the Supplementary Information. The experimental data have been deposited in our open-access repository: https://doi.org/10. 17877/RESOLV-2024-m057ffls. Any additional requests for information can be directed to, and will be fulfilled by, the corresponding authors.

## Code availability
Multimode is available as source code upon request to J.M.B., the program to compute VRT levels and spectra is available upon request to A.v.d.A.

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

## Acknowledgements

The authors thank Claude Leforestier for making available his computer programs based on the split Wigner pseudospectral method and for valuable discussions. This study has been funded by the Deutsche Forschungsgemeinschaft (DFG; German Research Foundation) under Germany's Excellence Strategy—EXC 2033—Projektnummer 390677874, NASA (80NSSC22K1167) and by the National Science Foundation (CHE-2154403). This publication is also based upon work of COST Action CA21101 on Confined molecular systems: from a new generation of materials to the stars (COSY) supported by COST (European Cooperation in Science and Technology).

## Author contributions

S.J., J.K., P.M. carried out the experiments; S.J., J.K., S.H., G.S., M.H. analyzed the experimental data and prepared figures; A.N. developed the potential and dipole moment surfaces, performed vibrational calculations, and prepared related figures; P.P. performed the diffusion Monte Carlo calculations and prepared related figures; K.R.B, M.A.P., G.S.T carried out the CCSD(T) computations and SAPT analysis; A.v.d.A. calculated the vibration-rotation-tunneling states and the far-infrared spectrum, prepared the corresponding figures and helped with the interpretation of the spectra; J.M.B., A.v.d.A., M.H. designed the study and wrote the paper.

## Funding

## Competing interests

The authors declare no competing interest.
