## [Peer Review File · Nature Communications]

On the nature of hydrogen bonding in the H₂S dimerReviewer #1 (Remarks to the Author):

The noteworthy results in the submission are the assignments of IR absorption peaks for three nuclear spin isomers of the H₂S dimer, (H₂S)₂. The results support the earlier assignment (Ref. 26 of the submission) of the microwave spectrum of the complex which established the fundamental geometry of the complex and confirmed the presence of a weak hydrogen bond. The findings of the present submission provide support for the overall model of the complex as provided in Ref. 26. It is weakly hydrogen bonded in a geometry similar to that of the water dimer with tunnelling splittings providing evidence of the floppiness of the complex. The new information is vibrational band assignments and the identification of nuclear spin isomers.

There are some places where the recorded spectra are described as “rotationally-resolved” which is not fully justified because the spectra displayed in the top of Figure 3 show extensively blended rotational features. The undulations of the broad bands are attributed to nuclear spin statistics in underlying rotational features. The evidence is broadly in favour of this interpretation but given the noise level and the linewidth, I think there is some doubt. Viewing the work holistically, however, the conclusions and claims presented are substantively supported by the data.

The experimental methodology is sound and the detail provided would allow methods to be reproduced.

Reviewer #2 (Remarks to the Author):

This manuscript reports (part of) the vibrational-rotational spectra of the H₂S dimer in a detail going beyond previous studies. Good agreement between high-resolution spectra and high quality quantum calculations enabled the detailed understanding of the studied, complicated band and assignment to vibrational-rotational-tunneling transitions. The Authors also present a comparison of the much better studied H₂O dimer to the case of H₂S from the perspectives of structure, interactions, quantum nuclear effects, vibrational spectra, etc.

The Authors employ highly advanced experimental and theoretical methods, although these approaches, separately, do not appear to go beyond the state-of-the-art. Their combined application and the so enabled understanding is valuable. Such high level of agreement and resolution is a great achievement, but not unprecedented at this range of system size and complexity.

The manuscript can be recommended for publication after revision according to the specific comments below. However, it is unclear if Nature Communications is the best suited journal for this work. The manuscript is highly technical and offers great value to the corresponding vibrational-rotational spectroscopy community. Although the Authors made efforts to write the Introduction and Discussion parts to a more general audience, the Results and other parts are hard to follow for non-experts. The only major concern against recommending for publication in Nature Communications is that, in the present form, it is not explained clearly what is the value and importance of the results for the broader readership of Nat. Commun. In the current version, it is not sufficiently explained why Nat. Commun. is better suited than a community

specific journal.

Specific comments:

Please, make much more clear what is the “new insight” (noted, etc. in the title and abstract) achieved, especially from the perspective of the broader audience. For example, the weaker electrostatics, more dispersion in the less directional interaction in H₂S dimer wrt. to H₂O dimer, and corresponding more delocalized and more strongly mixed modes are qualitative “insights”, which were probably noted in multiple studies from different perspectives before targeting S...H-S- type H-bonding. Reconciling the new and previous H₂S dimer spectra is a great and valuable achievement, but currently the broader impact of this result is not clearly communicated and not placed in the context of the S...H-S- type H-bond literature.

Could you explain in more detail to the non-expert reader how the nuclear spin is taken into account in the computations?

Could you explain in more detail to the non-expert reader what is the benefit of using He nanodroplets (compared to gas phase or matrix techniques)?

It would be very helpful for the analysis of Fig 1 and its comparison with the case of H₂O dimer if you could also add the H₂O dimer data to Fig 1. The side by side comparison of H₂O and H₂S on Fig 4 is much simpler to follow.

It is important and convincing to see the close agreement of the measurements in gas phase and He droplets for water dimer (p 9). However, considering the weaker interaction, higher delocalization and more small amplitude motions for the H₂S dimer, is it evident that the interaction with the He droplet is similarly negligible as for the H₂O dimer? Could you provide a more detailed argument, additional references or data supporting this assumption that the probably stronger dimer-droplet interaction for H₂S than for H₂O is still small enough?

The notation and discussion for the employed dimensions in the PES and quantum computations is confusing at some places when compared to each other: E.g., p11: “6+6D”, p15 “VSCF/VCI frequencies and intensities using the 12D PES...restricted to the six intramolecular modes”, p19 “12D VSCF/VCI calculations”...

How was the 10 cm⁻¹ uncertainty estimate determined on p 12?

It would be helpful to include numbers for the populations relevant at 0.37K and 4K for the analysis of the 2 investigated states as these numbers are probably not general knowledge.

P15: It would be helpful to add the noted VSCF/VCI numbers to Table 3 to make the comparison more simple for the reader. Then, it should be made more apparent which row of Table 3 is to be compared to which computed result.

p8, p15, p20 When noting agreement, please, be explicit about which experimental values agree with which computed numbers.

The major part of the discussion on pages 19 and 20 in a large extent repeats key points from the preceding Results section, which structure in the present form does not add as much value to the manuscript as it should. A Nat. Commun. reader would probably expect discussion on the broader impact of the specific Results in such a Discussion part.

Why were the same structure (bond lengths and angle) employed for both of the H₂S monomers in the rigid-monomer computations (p 22)? One may expect that a different structure for the donor and acceptor H₂S molecules could be more realistic.

Typos and text quality:

Unfortunately, especially compared to the large number of Authors (13) and their extensive scientific experience, there are a considerable number of typos or not clearly phrased parts requiring improvements. Additionally, it seems that the manuscript is written by multiple persons in multiple styles, which should be made more homogeneous, at least when these inconsistencies may cause confusion in the reader.

- The current text is difficult to read at places where there is only a note pointing to additional information located in an unspecified part of the SI. It would be very helpful to introduce section numbers in the SI and direct the reader to the precise place in the SI. (e.g., p 11, 13, 22-24...)

- p3: "ice: About"

- p5: verb seems to be missing: "the spectral overlap of the IR bands of H₂S clusters (dimers, trimers, or oligomers) in low-temperature matrices."

- p9 last paragraph: long assignment sentence would be easier to follow in a table format

- p13: 3/3/

- p13: one before last line: is Table 4 should be Figure 4? this part is unclear

- p14: cm⁻¹ italicized

- not consistent writing (Fig. vs Figure, dot and comma for decimals, SI or Supporting Information or Supplementary information...)

- p14: "As explained..." sentence is long and hard to follow.

- p14: "We want to point out ... experimental uncertainty." basically repeats the first paragraph on p9. Or, I miss what is new here and why this information has to be repeated.

- p15: "frequencies are ... are seen."

- p22: CCSD(T)-F12b/haQZ-F12 instead of CCSD(T) ?

- title page: "* both contributed equally" but there are 5 named labeled with *

Reviewer #3 (Remarks to the Author):

This paper reports a combined experimental and theoretical analysis the low-temperature spectrum of the H₂S dimer. It represents a very significant advance in the field, developing both a good spectroscopic model which explain both present and previous measurements, and highlighting significant differences between the H₂S and H₂O dimers. The paper is very well written and can be published once the authors have addressed some relatively minor issues:

1. The statement on page 3 "This has triggered a considerable growth in studies on the H₂S

molecule" would benefit from at least one reference supporting the statement. It is not statement I particularly recognize as being correct.

2. The column heading "gas phase" in Table 3 is somewhat misleading. It is pretty much impossible to obtain a standard gas phase (ie thermal equilibrium) high resolution spectrum of the H₂S dimer features considered in the gas phase. These data were obtained in a molecular beam and should be labelled as such.

3. The VSCF/VCI section starts by say that MULTIMODE calculations are "not at the same level of rigor" as the other calculations. While true this statement is very vague and not particularly helpful to the reader. Elsewhere the authors have been careful to quantify errors; it would be useful if this statement could be accompanied by some numbers.

Response to the reviewers

We thank all reviewers for their careful reading and helpful comments, which we have used to improve the manuscript. We have addressed all your concerns, see our point to point reply below, and we hope that the paper will now be accepted for publication.

Our answers to the reviewers are marked in **red**, and the text added to or modified in the revised manuscript in **blue**. In the marked version of the revised manuscript the revised parts are also indicated in **blue**.

Reviewer 1

The noteworthy results in the submission are the assignments of IR absorption peaks for three nuclear spin isomers of the H₂S dimer, (H₂S)₂. The results support the earlier assignment (Ref. 26 of the submission) of the microwave spectrum of the complex which established the fundamental geometry of the complex and confirmed the presence of a weak hydrogen bond. The findings of the present submission provide support for the overall model of the complex as provided in Ref. 26. It is weakly hydrogen bonded in a geometry similar to that of the water dimer with tunnelling splittings providing evidence of the floppiness of the complex. The new information is vibrational band assignments and the identification of nuclear spin isomers.

There are some places where the recorded spectra are described as “rotationally-resolved” which is not fully justified because the spectra displayed in the top of Fig. 3 show extensively blended rotational features. The undulations of the broad bands are attributed to nuclear spin statistics in underlying rotational features. The evidence is broadly in favour of this interpretation but given the noise level and the linewidth, I think there is some doubt. Viewing the work holistically, however, the conclusions and claims presented are substantively supported by the data.

First let us emphasize that for each of the substructures the observed intensity modulation was highly reproducible. For all bands we clearly observe *P* and *R* branches. In order to address the concerns of the referee we have now quantified the statistical uncertainty in the assignment and compared the quality of fits assuming a Boltzmann distribution with equal spin statistical weights and with alternating spin statistical weights. We carefully tested the effect of choosing alternative spin statistical weights on the quality of the fit of the band at 2604.17 cm⁻¹. As a suitable measure we used χ^2 , i.e., the sum of the weighted squared deviations between measurement and fit model. The weights of the individual measurements were defined as the inverse squares of their statistical errors. The optimum fit was obtained when assuming a spin

statistical weight $J_{\text{even}}:J_{\text{odd}}$ of 3:6, which corresponds to assigning the signal to the ortho-ortho transition. This yielded a χ^2 of 1110. When assuming instead spin statistical weight ratios of 1:1 or 6:3 we obtained considerably larger values of χ^2 : 1380 and 1819, respectively. The corresponding fits are shown in Fig. 1 of this response letter. Based on these results we assigned the band at 2604.17 cm^{-1} to the vibrational band of the ortho-ortho dimer.

The band at 2602.0 cm^{-1} showed no intensity alternations and followed a Boltzmann distribution, in agreement with an assignment to a vibrational transition of the ortho-para dimer (E symmetry) The third band at 2598 cm^{-1} also showed a particular feature: the spacing between the nearest lines of the P and R branches was found to be 0.14 cm^{-1} , which is significantly larger than for the other two bands where it is 0.09 cm^{-1} and 0.10 cm^{-1} . This spacing would be $4B$ if this band were assigned to E or B_2/A_2 symmetry and $6B$ for an assignment to A_1/B_1 symmetry, with B being the rotational constant. Thus, we assigned this band to a vibrational transition of the para-para dimer with spin statistical weights of 1 and 0 for alternating J levels. This clearly provides additional experimental evidence for our assignment. We have inserted the following text into the revised article:

To ensure an unambiguous assignment of the IR spectra we tested different spin statistical weight ratios of the rotational lines in the P and R branches of the band at 2604.2 cm^{-1} . It was found that a spin statistical weight ratio of 3:6 for $J_{\text{even}}:J_{\text{odd}}$ yields the smallest mean square deviation between measurement and fit. As explained in the last paragraph of the section “Theoretical far-infrared spectrum”, this intensity ratio of 3:6 indicates that this band belongs to $\text{oH}_2\text{S}-\text{oH}_2\text{S}$, with the lower states in the transitions having B_2^- symmetry for even J and A_2^+ symmetry for odd J . The fitted rotational constant of this band is $B = (0.0259 \pm 0.0003) \text{ cm}^{-1}$ and the difference between the vibrational ground and excited state values is $\Delta B = (-0.0009 \pm 0.0001) \text{ cm}^{-1}$. In the band at 2602.0 cm^{-1} no intensity alternation was observed. This agrees with an intensity ratio of 3:3, indicating that this band belongs to $\text{oH}_2\text{S}-\text{pH}_2\text{S}$, with lower states of E^+ symmetry for even J and E^- symmetry for odd J . The fitted rotational constants of this band are $B = (0.02230 \pm 0.0001) \text{ cm}^{-1}$ and $\Delta B = (-0.00067 \pm 0.00005) \text{ cm}^{-1}$. No unambiguous fit of the band at 2598.2 cm^{-1} was possible due to the low signal-to-noise ratio. The nearest lines of the P and R branches of this band are separated by 0.14 cm^{-1} , while for the other two bands, we find gaps of 0.09 and 0.10 cm^{-1} . This is in agreement with an assignment of the 2598.2 cm^{-1} band to $\text{pH}_2\text{S}-\text{pH}_2\text{S}$, with lower levels of A_1 symmetry for even J and B_1 symmetry for odd J . Since B_1 states have spin statistical weight zero, see Table 5, the odd J levels are missing. So the spacing should be $6B$, while for the ortho-ortho and ortho-para dimers it is $4B$, which nicely agrees with the observed gaps and confirms our assignment.

The experimental methodology is sound and the detail provided would allow methods to be reproduced.

Figure 1: Rotationally resolved band at 2604.17 cm^{-1} fitted to a Boltzmann distribution with different spin statistical weight ratios. The black fit at the bottom shows the original fit with a weight ratio $J_{\text{even}}:J_{\text{odd}}$ of 3:6. The purple fit in the middle of the graphic shows an equal statistical weight $J_{\text{even}}:J_{\text{odd}}$ of 1:1 and the top figure shows the fit with spin statistical weights $J_{\text{even}}:J_{\text{odd}}$ of 6:3.

Reviewer 2

This manuscript reports (part of) the vibrational-rotational spectra of the H_2S dimer in a detail going beyond previous studies. Good agreement between high-resolution spectra and high quality quantum calculations enabled the detailed understanding of the studied, complicated band and assignment to vibrational-rotational-tunneling transitions. The Authors also present a comparison of the much better studied H_2O dimer to the case of H_2S from the perspectives of structure, interactions, quantum nuclear effects, vibrational spectra, etc.

The Authors employ highly advanced experimental and theoretical methods, although these approaches, separately, do not appear to go beyond the state-of-the-art. Their combined application and the so enabled understanding is

valuable. Such high level of agreement and resolution is a great achievement, but not unprecedented at this range of system size and complexity.

The manuscript can be recommended for publication after revision according to the specific comments below. However, it is unclear if Nature Communications is the best suited journal for this work. The manuscript is highly technical and offers great value to the corresponding vibrational-rotational spectroscopy community. Although the Authors made efforts to write the Introduction and Discussion parts to a more general audience, the Results and other parts are hard to follow for non-experts. The only major concern against recommending for publication in Nature Communications is that, in the present form, it is not explained clearly what is the value and importance of the results for the broader readership of Nat. Commun. In the current version, it is not sufficiently explained why Nat. Commun. is better suited than a community specific journal.

Specific comments:

Please, make much more clear what is the “new insight” (noted, etc. in the title and abstract) achieved, especially from the perspective of the broader audience. For example, the weaker electrostatics, more dispersion in the less directional interaction in H₂S dimer wrt. to H₂O dimer, and corresponding more delocalized and more strongly mixed modes are qualitative “insights”, which were probably noted in multiple studies from different perspectives before targeting S...H-S-type H-bonding. Reconciling the new and previous H₂S dimer spectra is a great and valuable achievement, but currently the broader impact of this result is not clearly communicated and not placed in the context of the S...H-S- type H-bond literature.

We completely rewrote the conclusion, with emphasis on new insights and their broader interest. The new text reads:

Conclusion

The question whether the H₂S dimer is hydrogen bonded and how its properties compare with those of its smaller sibling, the H₂O dimer, attracted the interest of a broad chemical community^[1,2]. It triggered several infrared spectroscopic studies of the H₂S dimer in molecular beams^[3] and in various rare gas and other matrices.^[4,5,6,7,8] The spectra showed several partly overlapping bands in the frequency range of the S-H stretch mode, but their assignment remained uncertain due to the low resolution, the possible effects of different matrix sites, and the possibility that some of the bands belonged to trimers and larger clusters.

The microwave spectrum of the H₂S dimer measured in a molecular beam^[9] revealed clearly that this dimer has a hydrogen-bonded geometry, which evoked much attention.^[1,2] A hydrogen bonded geometry was also found in subsequent *ab initio* calculations of the dimer’s equilibrium structure.^[10]

The vast amount of studies of the H₂O dimer not only addressed its structure, but also the different tunneling processes connecting eight equivalent equilibrium structures and the intermolecular vibrations. High-resolution far-infrared spectra^[11,12,13,14,15,16], in combination with accurate calculations of its VRT states, yielded interesting information on the tunneling dynamics and the low-frequency intermolecular vibrations, which served as a very critical test of the available *ab initio* water pair potentials.^[17,18,19,20,21,22,23,24] Such information was still lacking for the H₂S dimer, but it is provided here. We present mass-selected IR spectra of the H₂S dimer in helium nanodroplets in a molecular beam, accompanied by calculations of its VRT states with the use of a new accurate *ab initio* pair potential and dipole function. The low temperature of 0.37 K and the superfluid nature of the helium droplets made it possible to resolve and analyse the rotational substructure of the observed infrared bands and, with the help of the calculated tunneling splittings in the vibrational ground state and the symmetry relations, to unambiguously assign the three bands in the measured spectrum. All three experimentally observed dimer bands are attributed to the stretch mode of the donor SH group directly involved in the hydrogen bond; this assignment is supported by VSCF/VCI calculations on the intramolecular vibrations which show that only this mode has sufficient intensity to be observed. Just as H₂O, H₂S has two nuclear spin isomers, ortho and para, which have different rotational states. The splitting of the donor SH stretch transition into three distinct bands is caused by different tunneling splittings in ortho-ortho, ortho-para, and para-para H₂S dimers. Also the different dissociation energies D_0 of the ortho-ortho, ortho-para, and para-para H₂S dimers into the corresponding monomers are reported here. The microwave spectrum^[9] only addressed the mixed ortho-para dimer, because it is the only dimer species with a non-vanishing dipole moment. The rotational constants calculated for the lowest two tunneling states of this ortho-para dimer agree well with the values deduced from the microwave spectrum^[9], which confirms that our *ab initio* H₂S dimer potential is accurate.

The calculations presented provide, for the first time, a full 12-dimensional potential and dipole function of the H₂S dimer depending on all intra- and intermolecular degrees of freedom. DMC calculations on this 12D potential yield the vibrational ground state of the dimer, while rigorous variational quantum calculations on the 6-dimensional intermolecular potential yield the tunneling and low-frequency intermolecular vibrational states. Moreover, we predict the H₂S dimer’s far-infrared spectrum, with the use of the 6D dipole function.

The tunneling splitting pattern in the H₂S dimer is found to be very different from the well-known pattern in the H₂O dimer, which is mainly caused by

more facile donor-acceptor interchange tunneling and more strongly hindered acceptor switch tunneling. Also the intermolecular vibrations differ strongly from those of the H₂O dimer, which is reflected by the predicted far-infrared spectrum. They have much lower frequencies, larger amplitudes, and different intermolecular modes are more strongly mixed. All of this indicates that the H₂S dimer is much floppier than the H₂O dimer. Calculations by symmetry-adapted perturbation theory, see Table S7 in the Supplementary Information, indicate that the less directional character of the hydrogen bond in the H₂S dimer, as compared to the H₂O dimer, is caused by a less dominant role of the electrostatic dipole and quadrupole interactions and relatively larger dispersion interactions, associated with the larger polarizability of H₂S. In summary, we conclude that the experimental spectra and their analysis, and the calculations presented in this paper clearly show that the H₂S dimer, although hydrogen bonded, shows very different dynamical properties than the isovalent H₂O dimer. The insights gained in this study of the H₂S dimer may also lead to a better understanding of the nature of non-covalent bonds involving sulfhydryl (SH) groups in biological systems, as compared to bonds involving OH groups.

Could you explain in more detail to the non-expert reader how the nuclear spin is taken into account in the computations?

The nuclear spin is not explicitly included in the calculations, but H₂S (just as H₂O) has two different nuclear spin states with total nuclear spin $I = 0$ and $I = 1$, accompanied by different rotational states. These are the so-called nuclear spin isomers called para and ortho H₂S. This is explained in the Methods section referring to the VRT state calculations and summarized in Table 5. We added the following text on the existence of para and ortho H₂S to the Introduction, where the ortho and para species are mentioned first:

As explained in the Methods section, H₂S (just as H₂O) has two different nuclear spin isomers, called para and ortho H₂S (pH₂S and oH₂S), which have different rotational states. Based on the observed nuclear spin multiplicities of alternating rotational levels, see Table 5, ...

Could you explain in more detail to the non-expert reader what is the benefit of using He nanodroplets (compared to gas phase or matrix techniques)?

The benefit of the use of helium nanodroplets rather than matrix techniques is clearly a much smaller frequency shift of the vibrational bands. Helium droplets are a very soft matrix with a matrix shifts on the order of 1 cm⁻¹. Also, no other matrix effects such as splittings by distinct sites are expected, due to the superfluid nature of the helium nanodroplets. A further advantage of the superfluidity is the conservation of the rotational substructure of the bands, although the separation between rotational lines is smaller than in the gas phase. An advantage of our experimental setup compared to the gas phase is the mass selectivity. This allowed us to unambiguously assign all three bands to the dimer (and other bands to larger clusters).

These benefits were already mentioned, but are now expressed somewhat more explicitly in various places.

It would be very helpful for the analysis of Fig 1 and its comparison with the case of H₂O dimer if you could also add the H₂O dimer data to Fig 1. The side by side comparison of H₂O and H₂S on Fig 4 is much simpler to follow.

We added the tunneling energy levels of H₂O-H₂O to Fig. 1 with the H₂S-H₂S levels.

See the new Fig. 1 in the revised manuscript.

It is important and convincing to see the close agreement of the measurements in gas phase and He droplets for water dimer (p 9). However, considering the weaker interaction, higher delocalization and more small amplitude motions for the H₂S dimer, is it evident that the interaction with the He droplet is similarly negligible as for the H₂O dimer? Could you provide a more detailed argument, additional references or data supporting this assumption that the probably stronger dimer-droplet interaction for H₂S than for H₂O is still small enough?

The main effect of the superfluid helium environment is the reduction of the dimer rotational constant by about a factor of 2.3, which is also observed for other molecules and dimers. Its effect on the vibrational frequencies is much smaller, mostly on the order of a few cm⁻¹. As one may observe by comparison of the gas phase (actually molecular beam) frequencies in Table 4 with our values observed in helium nanodroplets, this is also the case for H₂S-H₂S. In particular, the frequency of 2605 cm⁻¹ in the gas phase may be compared with the frequencies of 2598.2, 2602.0, and 2604.2 cm⁻¹ of the bands observed in helium nanodroplets. If one takes into account that our bands are assigned to the para-para, ortho-para, and ortho-ortho H₂S dimers and that their frequencies should be weighted with the spin statistical weights of 1, 3, and 9, respectively, to compare them with the single gas phase band the difference between the vibrational frequency in the gas phase and in helium droplets is less than 2 cm⁻¹. The gas phase frequency of 2590 cm⁻¹ in Table 3 agrees nicely with our value of 2588.4 cm⁻¹ observed in helium droplets for H₂S trimers. For the HCl-HCl complex, comparable in size with H₂S-H₂S, the frequency shift in helium droplets relative to the gas phase was found^[25] to be 5 cm⁻¹.

We reorganized Table 4 such that the frequencies observed in molecular beams and our helium droplet values can be directly compared.

See the new Table 4.

We also modified the text on the comparison of the helium droplet and gas phase frequencies, as follows:

It is well known and illustrated by our results below that the vibrational

frequencies of molecular dimers in He nanodroplets agree with the gas phase values to within a few cm^{-1} .

The notation and discussion for the employed dimensions in the PES and quantum computations is confusing at some places when compared to each other: E.g., p11: “6+6D”, p15 “VSCF/VCI frequencies and intensities using the 12D PES...restricted to the six intramolecular modes”, p19 “12D VSCF/VCI calculations”...

The symbols 6D and 12D are now defined as 6-dimensional (6D) and 12-dimensional (12D) when they occur for the first time. The symbol 6+6D has been omitted, because the meaning of this symbol is clearly explained in Ref. 15 which is cited. VSCF/VCI is defined as Vibrational Self-Consistent Field and Configuration Interaction (VSCF/VCI) when it occurs first.

See the corresponding additions to the article, also marked in blue.

How was the 10 cm^{-1} uncertainty estimate determined on p 12?

The diffusion Monte Carlo (DMC) method is a statistical sampling method and this number is based on the statistical uncertainty in the calculations of the dimer zero-point energy (ZPE) and an estimate of the inherent uncertainties in the 12-dimensional potential surface used to get the “exact” dimer and monomer ZPE’s. We added:

statistical

It would be helpful to include numbers for the populations relevant at 0.37K and 4K for the analysis of the 2 investigated states as these numbers are probably not general knowledge.

Assuming a Boltzmann distribution the populations of the lowest E^+ and E^- states separated by 1.23 cm^{-1} are 0.61 and 0.39 at the estimated temperature of 4K in a molecular beam, at the temperature of 0.37 K in helium nanodroplets these are 0.992 and 0.008. In the revised manuscript we rephrased the paragraph as follows:

The first indications for feasible tunneling were found in the microwave spectrum of the H_2S dimer measured by Arunan *et al.*^[26] in a molecular beam. They reported rotational transitions from two distinct ground states. These two states are the E^+ and E^- states, which we predict to be separated by 1.23 cm^{-1} . Using a Boltzmann distribution it can be estimated that at the molecular beam temperature of 4K the populations of the E^+ and E^- levels are 0.61 and 0.39, whereas at the superfluid helium droplet temperature of 0.37 K these are 0.992 and 0.008. Therefore, only the E^+ state is expected to be populated in superfluid helium droplets.

P15: It would be helpful to add the noted VSCF/VCI numbers to Table 3 to

make the comparison more simple for the reader. Then, it should be made more apparent which row of Table 3 is to be compared to which computed result.

We added the VSCF/VCI results to the caption of Table 4 (previously Table 3). The frequencies from molecular beam and matrix experiments were originally assigned to multiple bands (donor stretch, symmetric acceptor, free donor, and asymmetric acceptor stretch). We included both our new assignment of the bands in helium droplets and the original assignment of the bands in molecular beams in Table 4.

See the new Table 4 (previously Table 3) in the revised manuscript.

p8, p15, p20 When noting agreement, please, be explicit about which experimental values agree with which computed numbers.

On page 8 we added the maximum deviation of our calculated rotational constants from the experimental values^[9]. The explicit comparison no longer appears on page 20, because the Discussion and Conclusion section on pages 19 and 20 has been rewritten to avoid repetitions and to emphasize the relevance of our results, as reviewer 2 requested. On page 8 we added:

The maximum deviation of our calculated rotational constants from the values extracted from the microwave spectrum^[9] is 1.27%. This confirms the accuracy of the H₂S-H₂S potential used in our VRT calculations.

Page 15: This probably refers to the sentence “It is notable that the resulting splitting pattern in the energy schema of the excited state of H₂S dimer resembles remarkably well the tunneling splitting pattern in the ground state of water dimer.” This is just a qualitative statement and does not refer to a comparison between experiment and theory, so we cannot make it more explicit.

The major part of the discussion on pages 19 and 20 in a large extent repeats key points from the preceding Results section, which structure in the present form does not add as much value to the manuscript as it should. A Nat. Commun. reader would probably expect discussion on the broader impact of the specific Results in such a Discussion part.

The Discussion and Conclusion section was completely removed and replaced by a Conclusion that focuses more on the new insights obtained and their general interest.

The new Conclusion is reproduced in our answer to the first question of Reviewer 2, see above.

Why were the same structure (bond lengths and angle) employed for both of the H₂S monomers in the rigid-monomer computations (p 22)? One may expect that a different structure for the donor and acceptor H₂S molecules could be more realistic.

When considering a single dimer equilibrium structure the geometries of the donor and acceptor H₂S molecules are indeed different. However, there are eight equivalent equilibrium structures of the H₂S dimer, which are related by donor-acceptor interchange and other permutations. The H₂S dimer is floppy, its vibrational wave functions are strongly delocalized and include tunneling between all equivalent structures. In order to correctly describe all of these quantum motions the monomers must have the same (symmetric) geometries. This is a standard assumption in rigid monomer calculations on Van der Waals and hydrogen bonded dimers and it has been shown on the water dimer, for example, that such a rigid-monomer treatment accurately reproduces the tunneling and intermolecular vibrational levels and spectra calculated with flexible monomers. After the specification of the rigid-monomer geometry, the following text has been added:

It must be the same for both monomers in order to reproduce all tunneling effects including donor-acceptor interchange.

Typos and text quality:

Unfortunately, especially compared to the large number of Authors (13) and their extensive scientific experience, there are a considerable number of typos or not clearly phrased parts requiring improvements.

Additionally, it seems that the manuscript is written by multiple persons in multiple styles, which should be made more homogeneous, at least when these inconsistencies may cause confusion in the reader.

We carefully checked the whole manuscript, made the style more uniform, and made corrections where necessary. Specific places are mentioned below where we address the individual points of the reviewer.

The current text is difficult to read at places where there is only a note pointing to additional information located in an unspecified part of the SI. It would be very helpful to introduce section numbers in the SI and direct the reader to the precise place in the SI. (e.g., p 11, 13, 22-24...)

The sections in the Supplementary Information are now numbered and we specify explicitly to which sections, tables, and figures we refer.

We added section and table numbers in all references to the Supplementary Information where they were missing. These additions are indicated in blue in the revised manuscript.

p3: "ice: About" We merged the two sentences, writing:

The pair potential is the leading term in a many-body potential for liquid water and ice and accounts for 80-90% of the interactions in the bulk phase,

which triggered the extensive search for precise intermolecular potential energy surfaces.

p5: verb seems to be missing: “the spectral overlap of the IR bands of H₂S clusters (dimers, trimers, or oligomers) in low-temperature matrices.”

We reread the sentence and we think it is complete. The second part starts with “and” and it is a continuation of the “due to” in the first part. We removed the comma to make this more clear.

p9 last paragraph: long assignment sentence would be easier to follow in a table format

The assignments in this paragraph were replaced by a table that specifies these assignments (Table 2). The new paragraph, including the table, reads:

The lines in the spectrum of Fig. 3 were assigned with the use of the transition frequencies and line strengths calculated for each of the allowed irreps with the analytical method. The assignments are indicated in Fig. 3 and are listed in Table 2.

Table 1: Assignment of the lines in the theoretical far-infrared spectrum at $T = 0.37$ K. The symmetry labels are defined in Table 5. States with symmetry B_1^- have weight zero, so these odd J states are missing.

frequency cm ⁻¹	lower state		nature of excited state
	even J	odd J	
7.5	B_2^-	A_2^+	acceptor tunneling
13	A_1^+	B_1^-	interchange tunneling
22	E^+	E^-	{ donor stretch in-plane bend
26	B_2^-	A_2^+	{ donor torsion out-of-plane bend
32	$E^+(K=0)$	$E^-(K=0)$	$E^-, E^+(K=1)$
37	various	various	mixed

p13: 3/3/

The second slash was removed.

p13: one before last line: is Table 4 should be Figure 4? this part is unclear.

We rephrased the sentence to make it more clear. Now it reads:

While in previous studies^[3,27,8,4,6] the IR bands observed in molecular beams

and in matrices were assigned to the ν_1 donor/acceptor and ν_3 donor/acceptor modes, all observed H₂S dimer bands are now assigned to ro-vibrational tunneling transitions involving the bound donor SH stretch mode, see Table 4.

p14: cm-1 italicized

italics were replaced by roman font.

not consistent writing (Fig. vs Figure, dot and comma for decimals, SI or Supplementary information...)

The writing was made uniform by using Fig. and Supplementary Information (SI) throughout the manuscript. All decimals are now dots.

p14: "As explained": sentence is long and hard to follow.

This sentence appeared in text that was modified to meet the request of reviewer 1. In the new text, shown in our reply to reviewer 1, this sentence was split.

See the text in blue in our reply to reviewer 1.

p14: "We want to point out ... experimental uncertainty." basically repeats the first paragraph on p9. Or, I miss what is new here and why this information has to be repeated.

In our new text on pages 14 and 15 written to meet the request of reviewer 1 this text does not appear anymore. So the statement on page 9 is not repeated.

p15: "frequencies are ... are seen."

The second "are" was removed.

CCSD(T)-F12b/haQZ-F12 instead of CCSD(T) ?

CCSD(T) was changed into CCSD(T)-F12b/haQZ-F12

title page: "* both contributed equally" but there are 5 named labeled with *.

The * was removed from the first two authors and used to mark the corresponding authors. In the footnote it is mentioned the first two authors contributed equally.

Reviewer 3

This paper reports a combined experimental and theoretical analysis of the low-temperature spectrum of the H₂S dimer. It represents a very significant advance in the field, developing both a good spectroscopic model which explain both present and previous measurements, and highlighting significant differences

between the H₂S and H₂O dimers. The paper is very well written and can be published once the authors have addressed some relatively minor issues:

1. The statement on page 3 "This has triggered a considerable growth in studies on the H₂S molecule" would benefit from at least one reference supporting the statement. It is not statement I particularly recognize as being correct.

Several references have been added and the text has been rephrased, as follows:

Sulfhydryl (SH) compounds are well known for their weak non-covalent interactions. Cysteine, for instance, plays a role in establishing side chain conformations and stability of secondary structures in peptides through inter- and intramolecular interactions.^[28,29,30] The simplest SH-containing compound, hydrogen sulfide (H₂S), acts as a biological signaling molecule^[31,32,33] and as a superconductor precursor.^[34,35] H₂S is the simplest sulfur-bearing molecule detected in the interstellar medium (ISM) and plays an important role in astrochemistry.^[36,37]

2. The column heading "gas phase" in Table 3 is somewhat misleading. It is pretty much impossible to obtain a standard gas phase (ie thermal equilibrium) high resolution spectrum of the H₂S dimer features considered in the gas phase. These data were obtained in a molecular beam and should be labelled as such.

"Gas phase" in the header of Table 4 (previously Tabel 3) was replaced by "molecular beams". In the text we wrote:

molecular beams^[38,3]

We also rewrote the next paragraph, as follows:

Here, we report an IR spectrum of H₂S dimers in superfluid helium droplets in the frequency range of the bound S-H stretch mode. We were able to resolve a rotational structure and to assign the three bands observed at 2598.2, 2602.0, and 2604.2 cm⁻¹ to a single vibrational band with resolved tunneling states belonging to the ortho-ortho, ortho-para, and para-para nuclear spin isomers of the H₂S dimer. This is in line with our calculations, which predict well separated energy levels for ortho-ortho, ortho-para, and para-para H₂S dimers, but only a single strong vibrational transition in this frequency range. Bands observed at 2588 and 2620 cm⁻¹ were assigned to the trimer, based on the pick-up curves. Thus, we reassign the previously observed bands at 2590 and 2618 cm⁻¹ in molecular beams also to the trimer and the band at 2605 cm⁻¹ to the dimer. The reassignment of these bands is summarized in Table 4.

3. The VSCF/VCI section starts by say that MULTIMODE calculations are "not at the same level of rigor" as the other calculations. While true this statement is very vague and not particularly helpful to the reader. Elsewhere the authors have been careful to quantify errors; it would be useful if this statement

could be accompanied by some numbers.

It is not possible to quantify the errors due to the assumptions made in the VSCF/VCI calculations. But we extended the explanation of “not at the same level of rigor” in the following sentence by writing:

They do not take into account the delocalized nature of the VRT states, but instead they are done at three reference configurations which are nearly isoenergetic. Moreover, they do not include rotation of the dimer ($J = 0$).

References

- [1] Sam Lemonick. H₂S dimer forms hydrogen bonds. *Chem. Eng. News*, 96 (41), 2018. URL <https://cen.acs.org/physical-chemistry/chemical-bonding/H2S-dimer-forms-hydrogen-bonds/>
- [2] Katrina Krämer. Hydrogen sulfide surprises as it's discovered to have hydrogen bonds. <https://www.chemistryworld.com/news/hydrogen-sulfide-surprises-as-its-discovered-to-have-hydrogen-bonds/> 2018.
- [3] Aditi Bhattacharjee, Yoshiyuki Matsuda, Asuka Fujii, and Sanjay Wategaonkar. The intermolecular s-h-y (y=s,o) hydrogen bond in the h₂s dimer and the h₂s-meoh complex. *ChemPhysChem*, 14(5):905–914, 2013. ISSN 1439-4235. doi: <https://doi.org/10.1002/cphc.201201012>. URL <https://doi.org/10.1002/cphc.201201012>.
- [4] Hideji Tsujii, Kenji Takizawa, and Seiichiro Koda. Ir spectra of hydrogen bonding of h₂s doped in kr solids. *Chem. Phys.*, 285(2-3):319–326, 2002.
- [5] Esa Isoniemi, Mika Pettersson, Leonid Khriachtchev, Jan Lundell, and Markku Räsänen. Infrared spectroscopy of h₂s and sh in rare-gas matrixes. *J. Phys. Chem. A*, 103(6):679–685, 1999. ISSN 1089-5639. doi: [10.1021/jp9838893](https://doi.org/10.1021/jp9838893). URL <https://doi.org/10.1021/jp9838893>.
- [6] Anthony J. Tursi and Eugene R. Nixon. Infrared spectra of matrix-isolated hydrogen sulfide in solid nitrogen. *J. Chem. Phys.*, 53(2): 518–521, 1970. ISSN 0021-9606. doi: [10.1063/1.1674019](https://doi.org/10.1063/1.1674019). URL <https://doi.org/10.1063/1.1674019>.
- [7] Eric L. Woodbridge, Tai-Ly Tso, Mark P. McGrath, Warren J. Hehre, and Edward K.C. Lee. Infrared spectra of matrix-isolated monomeric and dimeric hydrogen sulfide in solid o₂. *J. Chem. Phys.*, 85(12):6991–6994, 1986.
- [8] A.J. Barnes and J.D.R. Howells. Infra-red cryogenic studies. part 7.—hydrogen sulphide in matrices. *J. Chem. Soc. Faraday Trans. 2*, 68:729–736, 1972.
- [9] A. Das, P. K. Mandal, F. J. Lovas, Ch. Medcraft, N. R. Walker, and E. Arunan. The h₂s dimer is hydrogen-bonded: direct confirmation from microwave spectroscopy. *Angew. Chem. Int. Ed.*, 57:15199–15203, 2018.
- [10] M. A. Perkins, K. R. Barlow, K. M. Dreux, and G. S. Tschumper. Anchoring the hydrogen sulfide dimer potential energy surface to juxtapose (h₂s)₂ with (h₂o)₂. *J. Chem. Phys.*, 152:214306, 2020. doi: [10.1063/5.0008929](https://doi.org/10.1063/5.0008929).
- [11] E. Zwart, J. J. ter Meulen, and W. L. Meerts. The submillimeter rotation tunneling spectrum of (d₂o)₂. *Chem. Phys. Lett.*, 173:115, 1990.

- [12] E. N. Karyakin, G. T. Fraser, and R. D. Suenram. Microwave spectrum of the $k_2 = 1 \leftarrow 0$ rotation-tunneling band of $(d_2o)_2$. *Mol. Phys.*, 78:1179, 1993.
- [13] J. B. Paul, R. A. Provencal, and R. J. Saykally. Characterization of the $(d_2o)_2$ hydrogen-bond-acceptor antisymmetric stretch by ir cavity ringdown laser absorption spectroscopy. *J. Phys. Chem. A*, 102:3279, 1998.
- [14] L. B. Braly, J. D. Cruzan, K. Liu, R. S. Fellers, and R. J. Saykally. Terahertz laser spectroscopy of the water dimer intermolecular vibrations i: $(d_2o)_2$. *J. Chem. Phys.*, 112:10293, 2000.
- [15] F. N. Keutsch, N. Goldman, H. A. Harker, C. Leforestier, and R. J. Saykally. Complete characterization of the water dimer vibrational ground state and testing the VRT(ASP-W)III, SAPT-5st, and VRT(MCY-5f) surfaces. *Mol. Phys.*, 101:3477–3492, 2003.
- [16] F. N. Keutsch, L. B. Braly, M. G. Brown, H. A. Harker, P. B. Petersen, C. Leforestier, and R. J. Saykally. Water dimer hydrogen bond stretch, donor torsion overtone, and “in-plane bend” vibrations. *J. Chem. Phys.*, 119:8927–8937, 2003.
- [17] G. C. Groenenboom, P. E. S. Wormer, A. van der Avoird, E. M. Mas, R. Bukowski, and K. Szalewicz. Water pair potential of near spectroscopic accuracy: II. vibration-rotation-tunneling levels of the water dimer. *J. Chem. Phys.*, 113:6702–6715, 2000. doi: 10.1063/1.1311290.
- [18] R. Bukowski, K. Szalewicz, G. C. Groenenboom, and A. van der Avoird. Interaction potential for water dimer from symmetry-adapted perturbation theory based on density functional description of monomers. *J. Chem. Phys.*, 125:044301, 2006. doi: 10.1063/1.2220040.
- [19] R. Bukowski, K. Szalewicz, G. C. Groenenboom, and A. van der Avoird. Predictions of the properties of water from first principles. *Science*, 315:1249–1252, 2007. doi: 10.1126/science.1136371. URL <http://www.sciencemag.org/cgi/content/abstract/315/5816/1249>.
- [20] R. Bukowski, K. Szalewicz, G. C. Groenenboom, and A. van der Avoird. Polarizable interaction potential for water from coupled cluster calculations. ii. applications to dimer spectra, virial coefficients, and simulations of liquid water. *J. Chem. Phys.*, 128:094314, 2008. doi: 10.1063/1.2832858.
- [21] X. Huang, B. J. Braams, J. M. Bowman, R. E. A. Kelly, J. Tennyson, G. C. Groenenboom, and A. van der Avoird. New *ab initio* potential energy surface and the vibration-rotation-tunneling levels of $(h_2o)_2$ and $(d_2o)_2$. *J. Chem. Phys.*, 128:034312, 2008. doi: 10.1063/1.2822115.
- [22] C. Leforestier, R. van Harrevelt, and A. van der Avoird. Vibration-rotation-tunneling levels of the water dimer from an *ab initio* potential surface

- with flexible monomers. *J. Phys. Chem. A*, 113:12285–12294, 2009. doi: 10.1021/jp9020257.
- [23] C. Leforestier, K. Szalewicz, and A. van der Avoird. Spectra of water dimer from a new *ab initio* potential with flexible monomers. *J. Chem. Phys.*, 137:014305, 2012. doi: 10.1063/1.4722338.
- [24] V. Babin, C. Leforestier, and F. Paesani. Development of a “first principles” water potential with flexible monomers: dimer potential energy surface, vrt spectrum, and second virial coefficient. *J. Chem. Theor. Comp.*, 9:5395–5403, 2013.
- [25] Markus Ortlieb and Martina Havenith. Proton Transfer in (HCOOH)₂: An IR High-Resolution Spectroscopic Study of the Antisymmetric C-O Stretch. *J. Phys. Chem. A*, 111:7355–7363, 2007.
- [26] Arijit Das, Pankaj K. Mandal, Frank J. Lovas, Chris Medcraft, Nicholas R. Walker, and Elangannan Arunan. The h₂s dimer is hydrogen-bonded: Direct confirmation from microwave spectroscopy. *Angew. Chem. Int. Ed.*, 57(46):15199–15203, 2018. ISSN 1433-7851. doi: <https://doi.org/10.1002/anie.201808162>. URL <https://doi.org/10.1002/anie.201808162>.
- [27] P. Soulard and B. Tremblay. Vibrational study in neon matrix of h₂s-h₂o, h₂s-(h₂o)₂, and (h₂s)₂-h₂o complexes. identification of the two isomers: Hoh-sh₂ (h₂o proton donor) and hsh-oh₂ (h₂s proton donor). *J. Chem. Phys.*, 151(12):124308, 2019. doi: 10.1063/1.5120572.
- [28] L. M. Gregoret, S. D. Rader, R. J. Fletterick, and F. E. Cohen. Hydrogen bonds involving sulfur atoms in proteins. *Proteins: Structure, Function, and Bioinformatics*, 9:99–107, 1991.
- [29] P. Zhou, F. Tian, F. Lv, and Z. Shang. Geometric characteristics of hydrogen bonds involving sulfur atoms in proteins. *Proteins: Structure, Function, and Bioinformatics*, 76:151–163, 2009.
- [30] W. M. Westler, I. J. Lin, A. Perczel, F. Weinhold, and J. L. Markley. Hyperfine-shifted ¹³c resonance assignments in an iron-sulfur protein with quantum chemical verification: Aliphatic ch· · · s 3-center 4-electron interactions. *J. Am. Chem. Soc.*, 133:1310–1316, 2011.
- [31] Martin N. Hughes, Miguel N. Centelles, and Kevin P. Moore. Making and working with hydrogen sulfide: The chemistry and generation of hydrogen sulfide in vitro and its measurement in vivo: A review. *Free Radical Biology and Medicine*, 47(10):1346–1353, 2009. ISSN 0891-5849. doi: <https://doi.org/10.1016/j.freeradbiomed.2009.09.018>. URL <https://www.sciencedirect.com/science/article/pii/S0891584909005449>.

- [32] Yi-Hong Liu, Ming Lu, Li-Fang Hu, Peter T. H. Wong, George D. Webb, and Jin-Song Bian. Hydrogen sulfide in the mammalian cardiovascular system. *Antioxidants & Redox Signaling*, 17(1):141–185, 2012. ISSN 1523-0864. doi: 10.1089/ars.2011.4005. URL <https://doi.org/10.1089/ars.2011.4005>.
- [33] Khosrow Kashfi and Kenneth R. Olson. Biology and therapeutic potential of hydrogen sulfide and hydrogen sulfide-releasing chimeras. *Biochemical Pharmacology*, 85(5):689–703, 2013. ISSN 0006-2952. doi: <https://doi.org/10.1016/j.bcp.2012.10.019>. URL <https://www.sciencedirect.com/science/article/pii/S000629521200696X>.
- [34] Y. Li, J. Hao, H. Liu, Y. Li, and Y. Ma. The metallization and superconductivity of superconductivity of dense hydrogen sulfide. *J. Chem. Phys.*, 140:174712, 2014.
- [35] A. P. Drozdov, M. I. Erements, I. A. Troyan, V. Ksenofontov, and S. I. Shylin. Conventional superconductivity at 203 kelvin at high pressures in the sulfur hydride system. *Nature*, 525:73–76, 2015.
- [36] Yasuhiro Oba, Takuto Tomaru, Akira Kouchi, and Naoki Watanabe. Physico-chemical behavior of hydrogen sulfide induced by reactions with h and d atoms on different types of ice surfaces at low temperature. *Astrophys. J.*, 874(2):124, 2019. ISSN 1538-4357. doi: 10.3847/1538-4357/ab0961. URL <http://dx.doi.org/10.3847/1538-4357/ab0961>.
- [37] Duncan V. Mifsud, Zuzana Kaňuchová, Péter Herczku, Sergio Ioppolo, Zoltán Juhász, Sándor T. S. Kovács, Nigel J. Mason, Robert W. McCullough, and Béla Sulik. Sulfur ice astrochemistry: A review of laboratory studies. *Space Science Reviews*, 217(1):14, 2021. ISSN 1572-9672. doi: 10.1007/s11214-021-00792-0. URL <https://doi.org/10.1007/s11214-021-00792-0>.
- [38] Sh. Sh. Nabiev, L. A. Palkina, and V. I. Starikov. Ir spectra and dipole moment function of the h₂s molecule in the gas and liquid phases. *Russ. J. Phys. Chem. B*, 7:721–733, 2013.

Reviewer #1 (Remarks to the Author):

The substantive issue raised in my previous review was:

"There are some places where the recorded spectra are described as "rotationally resolved" which is not fully justified because the spectra displayed in the top of Fig. 3 show extensively blended rotational features. The undulations of the broad bands are attributed to nuclear spin statistics in underlying rotational features. The evidence is broadly in favour of this interpretation but given the noise level and the linewidth, I think there is some doubt."

I am satisfied that the authors have addressed this point within the revision.

Reviewer #2 (Remarks to the Author):

The Authors invested considerable effort to improve the manuscript and responded in an acceptable manner to most of the comments of all three reviewers. The manuscript now can be recommended for publication.

Reviewer #3 (Remarks to the Author):

The authors have substantially re-written their paper in response to both my comments and those of the other referees. The paper is significantly improved and can be accepted for publication.